

# Segmented flow coil equilibrator coupled to a Proton Transfer Reaction Mass Spectrometer for measurements of a broad range of Volatile Organic Compounds in seawater

Charel Wohl[1,2], David Capelle[3], Anna Jones[4], William T. Sturges[2], Philip D. Nightingale[1,2], Brent G. T. Else[5],
Mingxi Yang[1]

[1]Plymouth Marine Laboratory, Plymouth, PL1 3DH, United Kingdom
[2]School of Environmental Sciences, University of East Anglia, Norwich NR4 7TJ, United Kingdom
[3]Center for Earth Observation Science, University of Manitoba, Winnipeg, Manitoba, Canada
[4]British Antarctic Survey, Cambridge, High Cross, Madingley Road, CB3 0ET, United Kingdom
[5]Department of Geography, University of Calgary, Calgary, T2N 1N4, Canada

*Correspondence to*: Mingxi Yang (miya@pml.ac.uk)

**Abstract.** Here we present a technique that utilises a segmented flow coil equilibrator coupled to a Proton Transfer Reaction-Mass Spectrometer to measure a broad range of dissolved organic gases. Due to its unique
design composed of a segmented flow and a headspace-water separator, the equilibrator is highly efficient for gas exchange and has a fast response time (under 1min). The system allows for both discrete and continuous measurements of volatile organic compounds in seawater due to its ease of changing sample intake and low sample water flow (100 cm$^3$ min$^{-1}$). The equilibrator setup is both relatively inexpensive and compact. Hence it can be easily reproduced and installed on a variety of oceanic platforms, particularly where space is limited. As
a result of its smooth and unreactive surfaces, the segmented flow coil equilibrator is expected to be less sensitive to biofouling and easier to clean than membrane-based equilibration systems. The equilibrator fully equilibrates for gases that are similarly soluble or more soluble than toluene. The method has been successfully deployed in the Canadian Arctic. Here, some example data of underway surface water and Niskin bottle measurements in the sea ice zone are presented to illustrate the efficacy of this measurement system.

## 1 Introduction

Volatile organic compounds (VOCs) are present ubiquitously throughout the atmosphere (Heald et al., 2008) and play important roles in ozone chemistry (Monks, 2005) and OH concentrations (Lewis et al., 2005) as well as particle formation (Blando & Turpin, 2000). For example, acetone, acetaldehyde and methanol particularly affect the oxidative capacity of the remote marine atmosphere (Lewis et al. 2005). The oxidation
products of dimethyl sulphide (DMS) and isoprene are important particle precursors in the marine atmosphere that affect cloud formation and the albedo effect of the planet (Charlson, Lovelock, Andreae, & Warren, 1987; Claeys et al., 2004). Benzene and toluene are organic pollutants emitted by ship traffic (Saeed & Al-Mutairi, 1999). The oceans play an important role in controlling atmospheric VOC concentrations by exchanging VOCs with the atmosphere (Carpenter et al. 2012).



Current estimates of air–sea VOC fluxes and the cycling of VOCs in the oceans have been limited in part by our ability to measure these compounds in the water phase. Only a small number of methods allow for in situ quantification of VOCs. For example, derivatisation methods have been used, which require the synthesis of toxic chemicals to determine aldehyde concentrations in seawater with detection by high performance liquid chromatography (Zhu & Kieber, 2018). Such methods are not suitable for measuring a large number of samples.

Most methods of detection require the analyte to be in the gas phase, necessitating an adequate extraction or equilibration device.

      Previous waterside measurements have primarily been made using purge and trap (PT) systems coupled to Gas Chromatograph – Mass Spectrometers (de Bruyn, Clark, Senstad, Barashy, & Hok, 2017). This is sensitive enough to allow detection in seawater (quantification down to nmol dm$^{-3}$) but requires manual handling and is

often more suitable for discrete measurements. A Gas Chromatograph - Mass Spectrometer has been coupled to a PT system to measure benzene and toluene amongst other compounds (Huybrechts, Dewulf, Moerman, & Langenhove, 2000). Others have coupled PT systems to a Gas Chromatograph - Flame Ionisation Detector to measure isoprene (Exton, Suggett, Steinke, & McGenity, 2012), ethanol, and propanol in seawater (Beale et al. 2010). However, these setups are only suitable for discrete samples with a sample treatment time of under 2h

and care must be taken to avoid wall adsorption and desorption effects in the setup. A ship-based PT Gas Chromatograph - Mass Spectrometer has been used to measure a broad range of VOCs in discrete surface water samples with a three-hour frequency. However this required two people and represented a considerable workload (Schlundt et al., 2017). Some purge and trap systems have been automated to allow for underway measurements of halocarbons, DMS, and isoprene semi-continuously every ca. 30 minutes (Andrews et al.

2015). Extracted or equilibrated air from seawater contains a large amount of water vapour, which potentially affects the VOC detection. Thus, a dryer is often used to reduce the humidity in the sample air.  Measurement of very reactive/sticky gases such as methanol or acetone is problematic using this method due to adsorption and desorption on the dryer or tubing (Beale, Liss, Dixon, & Nightingale, 2011; Kameyama et al., 2010). Additionally, the long measurement time does not enable high-resolution measurements required for more

reactive gases that display fine scale variability. This highlights the need for continuous and automated measurement techniques that do not require pre-treatment.

      Two types of equilibrators are commonly used for continuous measurements of dissolved gases. One type allows for direct exchange between the carrier gas and the water, while the other uses a membrane to extract gases. Directly exchanging equilibrators such as the Weiss-style showerhead equilibrator (Johnson,

1999) allow underway $CO_2$ measurements with a <35 minute frequency. This has been used widely to measure $CO_2$ and short lived halocarbons (Arévalo-Martínez et al., 2013; Butler et al., 2007). However, spray generated from the showerhead lengthens the equilibrator's response time for highly soluble gases, making it less suitable for high frequency measurements of highly soluble VOCs such as methanol (Kameyama et al., 2010). Membrane equilibrators avoid spray formation and allow for selective diffusion. They have previously been used to extract

volatile compounds from the water phase continuously (Tortell, 2005). By using a hydrophobic membrane, the amount of water vapor in the detector can be reduced. For example membrane inlet mass spectrometers have been used to measure DMS and inorganic gases in seawater with a measurement frequency of more than once per minute (Tortell, 2005). Underway measurements of seawater DMS concentrations have been made with a 1



minute frequency using a Chemical Ionisation Mass Spectrometer (CIMS) coupled to a porous Teflon membrane
(Saltzman, de Bruyn, Lawler, Marandino, & McCormick, 2009). One disadvantage of membrane equilibrators is
that the equilibration efficiency could be affected by biological growth on the membrane surface (biofouling),
especially in biologically productive areas where some VOCs are known to have strong sources.

The choice of detector that the equilibrator is coupled to is crucial as well. Proton Transfer Reaction -
Mass Spectrometry (PTR-MS) is a widely used tool that allows high-frequency (0.1–1s) measurement of a broad
range of trace gases in the atmosphere (Lindinger and Jordan 1998; Blake et al. 2009).  It is similarly suitable for
high-resolution ship-based measurements of VOCs. Efforts have been made to quantify methanol, acetone, and
acetaldehyde in discrete water samples using a membrane system coupled to PTR-MS (Beale et al., 2011). This
represents a significant advance over the methods described above as there is no need for sample pre-
treatment and the setup does not contain reactive surfaces. Others have used a purge and trap system coupled
to PTR-MS to measure four different VOCs at a time (Williams et al., 2004). A bubbling-type equilibrator has also
been developed for underway measurements of a range of dissolved VOCs with PTR-MS (Kameyama et al.,
2010). The large volume of the bubbling equilibrator (i.d. 15.2 cm, height 100 cm) makes it very bulky and
creates a long response time (about 10min). Moreover, the high-water flow requirement of this type of
equilibrator (1 dm$^3$ min$^{-1}$) is also less suitable for discrete measurements.

In this paper, we present a segmented flow coil equilibrator (SFCE) that is adopted from the designs
used by Xie et al. (2001) and Blomquist et al. (2017) for measurements of carbon monoxide and DMS,
respectively.  We couple this equilibrator to a PTR-MS with the settings optimised for measurement of VOCs in
the water phase. The main advantage of this equilibrator lies in its design. Briefly, the segmented flow allows for
a large surface area for gas exchange, ample equilibration time, and so a high degree of equilibration. The
simple headspace and water separation system allows for rapid drainage of the sampled water as well as
separation of the headspace from water without spray droplet formation. This reduces the response time to
below 1 min. Due to the ease of changing the water sample intake and low water flow (100 cm$^3$ min$^{-1}$), the
equilibrator can conveniently be used for both continuous underway and discrete measurements. The
equilibrator is entirely made up of commercially available Polytetrafluoroethylene (PTFE) tubing and fittings,
which should minimise adsorptive loss and make the equilibrator relatively inexpensive and easy to replicate.
The constant flow of water and smooth surfaces should also reduce bio fouling and facilitate occasional
cleaning. The equilibrator is described in detail in Sect. 2.1. A consideration in making seawater VOC
measurements is the difficulty in getting an accurate background, or blank. Hence the choice of dissolved VOC
background concentration is discussed and described in Sect. 3.1. The computation of dissolved and expected
VOC concentrations is described in Sect. 3.2. We assess the performance of the SFCE-PTR-MS system in Sect. 4,
focusing in particular on the equilibration efficiency and response time. The final consideration with using PTR-
MS for measurement of dissolved gases is the effect of humidity on the measurement calibration. A discussion
on how this is addressed can be found in Sect. 4.1 and in the appendix.



## 2 System description

### 2.1 Segmented flow coil equilibrator

The design of our SFCE is shown in Fig. 1. The SFCE is coupled to PTR-MS for measurement of methanol, acetone (2-propanone), acetaldehyde (ethanal), dimethyl sulphide (DMS), isoprene (2-methyl-1,3-butadiene), benzene and toluene (methyl benzene). These gases cover a large solubility range of solubilities (see Sect. 3.2). This demonstrates the versatility of the SFCE.

The SFCE takes approximately equal, continuous flows of high purity zero air (100 $cm^3$n $min^{-1}$ , where n indicates normalised mass flow at 0 °C, 1 atm; controlled by a Bronkhorst mass flow controller) and unfiltered seawater (95±5 $cm^3$ $min^{-1}$, controlled by a peristaltic pump, Watson Marlow 120 S/DV with 8 cm long Pumpsil platinum cured silicone tubing 4.4 mm i.d.). We used either Ultra-low VOC zero air (Praxair) scrubbed by a hydrocarbon trap or BTCA grade zero air (BOC) oxidized by a custom-made Platinum-catalyst (heated to 450°C) as the zero air carrier gas for the SFCE. Complete oxidation of VOCs in the custom made PT-Catalyst has been demonstrated previously for both dry air and air that is fully saturated with water at 20°C (Yang & Fleming, 2018).

The seawater is supplied either from the ship's underway water system or via 900 $cm^3$ glass sample bottles in the case of discrete measurements (Sect. 2.2). The carrier gas and water meet in a Teflon tee (4 mm inner diameter), which naturally leads to the formation of distinct segments of carrier gas and water. The segments travel through a coiled, 10 m long PTFE tube (outer diameter 6.35 mm, wall thickness, 1.19 mm). Each segment of carrier gas or water is approximately 1.5 cm long, giving an approximate total surface of exchange of 82 $cm^2$ in the coil. The coil is immersed in a water bath kept at a constant temperature of 20°C. The residence time in the 10 m tube is approx. 0.6 min. Laboratory measurements indicate that regardless of the initial water temperature (0-25°C), the water exiting the equilibrator has a temperature of 20±1°C. Keeping the temperature essentially constant has the benefit of (i) simplifying calibrations/calculations of aqueous concentration, and (ii) in the case cold, high latitude seawater samples, increasing the VOC signal in the headspace as warming to 20°C reduces the gas solubility. A rapid biological response to this warming is not expected due to the very short residence time in the segmented flow coil of 0.6 min.

In the initial design, after equilibration in the coiled segmented flow tube, the carrier gas-analyte mixture is separated in a 200 $cm^3$ PTFE jar (Savillex). Here, the gas mixture is going towards the PTR-MS and the seawater is drained away rapidly via a U-shaped drain (Fig. 1B). The U-shaped drain prevents intrusion of lab air and prevents the equilibrated gas from escaping through the water drain. We estimate a response time of about 2 minutes with the PTFE jar as the air-water separator. This is due to a combination of its sizable internal volume and the production of water droplets inside of the jar, which buffered the headspace to step-changes in seawater concentration in the case of highly soluble gases. After the Arctic field deployment, the PTFE jar was found to slightly outgas some VOCs (see Sect. 3.1 for further information) and was replaced with a PTFE tee piece (Swagelok, outer diameter: 12.7 mm, wall thickness 1.6 mm). This modification improved the system response time to less than 1 minute by greatly reducing the volume of the air-water separator and allowing for a smooth separation of the headspace and water segments without droplet formation (see Sect. 4.4). This also





allows for the entire SCFE to consist of readily available PTFE tubing and fittings routinely used for air sampling and are thus not expected to interact with these compounds. See Fig. 1 for these two designs.

150  At the exit of the equilibrator, the humid headspace air is diluted with dry zero air (20 $cm^3n$ $min^{-1}$, same as the carrier gas, controlled by another Bronkhorst mass flow controller).  This prevents condensation in the ~2 m PTFE tubing between the equilibrator and the heated (80°C) inlet of the PTR-MS (Fig. 1). This facilitates installation on board as detector and equilibrator can be within approx. 5m from each other. The SFCE system is operated at slight overpressure (measured to be approx. 0.024 mbar above atmospheric pressure) in order to reduce the likelihood of lab air contamination (e.g. due to leaks).  A vent is installed upstream of the PTR-MS to avoid pressurizing the detector. The vent flow is typically ~20 $cm^3$ $min^{-1}$ – the residual between the carrier gas
155  flow (100 $cm^3n$ $min^{-1}$), the dilution flow (20 $cm^3n$ $min^{-1}$), as well as the PTR-MS intake flow (~100 $cm^3$ $min^{-1}$).

 The entire SFCE system fits on a bench space of about 40 cm by 40 cm (see appendix). The SFCE is designed such that a failure of an individual component does not result in a catastrophic over or under-pressurization of the system. For example, if the carrier gas is stopped (e.g. gas supply runs out), the PTR-MS simply measures lab air via the vent and the water is drained from the SFCE as usual. If the water flow from the
160  underway sampling stops, the peristaltic pump will simply pump lab air into the equilibrator. These unexpected failures can be easily identified as lab air has typically much higher concentrations of VOCs than marine air or equilibrator headspace gas. If the PTR-MS fails, the headspace gas simply exits via the vent and/or the top of the U-shaped drain.

 Due to the smooth surfaces and constant and complete water renewal, the equilibrator should not be
165  very prone to biofouling. The lack of a membrane for gas exchange means that the degree of equilibration should not vary significantly even if there is some biofouling. To clean the SFCE if necessary, the seawater intake and the water drain pipe are connected to a 10% HCl solution for 10 min. During this operation, the PTR-MS is disconnected to sample lab air and the gas flow is stopped. A flow of HCl thus covers all the parts of the equilibrator that are normally exposed to seawater. To resume measurement of ambient seawater, the flow of
170  HCl is stopped and the carrier gas flow is started to drain the HCl safely into the recirculated solution. The equilibrator is typically rinsed with seawater before resuming measurement.

### 2.2 Field deployment

 The SFCE has been field tested on a three-week research cruise in the Arctic for measurements of underway surface water and discrete samples from depth profiles (Sect. 5). For underway measurements,
175  seawater from the ship seawater supply was continuously piped into an open-topped PTFE beaker fixed in the sink and allowed to overflow. The seawater was pulled by the peristaltic pump into the SFCE from the bottom of this beaker. This is to buffer pressure variations and so variable flow rates in the underway water supply, which could affect instrument response (Sect. 4.2). The open topped beaker also allows marine debris to diffuse and escape from the top, rather than clogging the SCFE intake. Since there are no membranes, small particles that
180  do enter the SCFE simply pass through the 4 mm inner diameter tube and are drained away.



Discrete water samples from the ship's rosette were collected in 900 cm³ ground stopper glass sample bottles via Tygon tubing. Sample bottles were rinsed three times and overfilled without introducing bubbles to avoid air contamination. To measure discrete samples, the underway measurement was stopped, and the PTFE water intake tube was simply moved from the seawater intake to each sample bottle (water flow stopped

during changeover). Water was pumped from the bottom of the 900 cm³ sample bottles, while minimizing agitation. The top 5 cm of the discrete water sample was not measured because of the possibility of air contamination. Sampling time per bottle was under 9 min. The analysis of about eight discrete samples including blank measurements (Sect. 3.1) is typically finished within two hours of sample collection. This should be fast enough to avoid sample degradation of even the most reactive VOCs (Beale et al., 2011). To verify that

the seawater supplied by the ship's underway water supply is uncontaminated, we sampled at every station from 5m depth. The underway and rosette samples are compared in Sect. 5.

**3 Derivations of dissolved VOC concentrations**

The PTR-MS measures VOC mixing ratios (in ppb) in the headspace of the equilibrator. Below we discuss how to convert these mixing ratios to dissolved waterside concentrations (in nmol dm⁻³). Headspace

equilibrator VOC mixing ratios are converted to ppb using the ideal gas law as stated in Eq. (1):

$$\frac{n}{V} = \frac{P}{R * T} \tag{1}$$

Where $n$ (mol) represents the quantity of matter, $V$ (dm³) represents the volume of gas, $P$ (Pa) represents the pressure, $R$ (m³ Pa K⁻¹ mol⁻¹) represents the gas constant of 8.314 and $T$ (K) represents the temperature in the segmented flow tube of 293.15 K. A conversion factor of 0.001 is applied to convert from m³ to dm³.

The degree of equilibration for each gas in the SCFE was determined experimentally and is presented in

Sect. 4.3. For compounds that fully equilibrate in the equilibrator, the following Eq. (2) is used to compute the measured dissolved gas concentrations:

$$C_w = \left(C_a - C_{a_o}\right) * H * PF * 1.2 \tag{2}$$

Where $C_w$ (nmol dm⁻³) represents the dissolved waterside concentration, $C_a$ (ppb) represents the measured headspace mixing ratio, $C_{a_o}$ (ppb) represents the background mixing ratio (see Sect. 3.1), $H$ represents the

dimensionless liquid-over-gas form of Henry solubility (see Sect. 3.2), $PF$ represents a purging factor (see appendix, text below), a factor of 1.2 is applied to account for the dilution of these gases in the headspace of the equilibrator.

Equilibrator headspace is laden with water vapour and humidity is known to influence the PTR-MS measurement (de Gouw & Warneke, 2007). For a more detailed discussion on the settings of the PTR-MS during

deployment, the computation of VOC mixing ratios in the PTR-MS, and the effects of humidity on the signal amplitude and background, please see the appendix.



For compounds that partially equilibrate, the mean calibration curve estimated from liquid standards diluted in MilliQ water (S in ppb nmol$^{-1}$ dm$^3$) was used to determine measured waterside concentrations after subtraction of the background;

$$C_w = \left(C_a - C_{a_o}\right) * \frac{1}{S}$$

(3)


This is more suitable than Eq. (2) as the Henry solubility and the purging factor do not apply for partially equilibrating gases. Technically, using a freshwater calibration curve to calculate gas concentrations in seawater will introduce an uncertainty (nominally within 20%) due to the effect of salinity on gas solubility. Of all the VOCs studied here, the highly insoluble isoprene is the only one that does not completely equilibrate in the
SFCE. The salting out effect of isoprene seems small relative to the uncertainty in the isoprene calibration curves (Sect. 3.2) and is thus neglected here.

### 3.1 Estimation of the background and system blank

The ideal system blank for seawater VOC measurements would be VOC-free seawater. However, we have been unable to generate or obtain seawater that is free of methanol, acetone or acetaldehyde due to the high
solubility and ubiquity of these gases. Additionally, it is debatable which seawater may be free of methanol, acetone or acetaldehyde due to a lack of understanding in the cycling of these compounds. The choice of background is most important for the soluble VOCs (methanol, acetone, acetaldehyde) as the ratio of background to signal can be quite high and the background can be variable. For example, for acetone the average signal to background ratio during field testing for three weeks was 1.62 with a background standard
deviation of 26%. Below, we discuss three different approaches at estimating the background of the seawater VOC measurements.

First, direct measurements of carrier gas zero air (i.e. bypassing the SFCE) are used to track any drift in the internal PTR-MS background. This simple method of blanking was used by e.g. Yang et al., (2013). Since humidity is known to affect the background of some of the measured compounds (de Gouw & Warneke, 2007)
(see appendix), we do not expect bypassing the equilibrator with zero air to give the most representative blanks for all seawater VOCs because (i) zero air has a much lower humidity than the equilibrator headspace; (ii) it does not account for any possible contamination within the equilibrator.

Second, at every sampling station, water from the bottom of the water column was collected and measured. Here we define the bottom water as the deepest water sample provided by the rosette, which was
between 290 m and 1700 m (well below the mixed layer). For certain VOCs that are only produced in the surface ocean and rapidly consumed at depth (such as DMS), it might be expected that their concentrations in deep water to be close to zero. However, there is insufficient field data to know whether this is the case for all the VOCs monitored here. Measurements of methanol and acetone in the north Atlantic show that their concentrations do decrease below the mixed layer (Williams et al., 2004) but do not necessarily go to zero.
Similarly, depth profile measurements showed acetone concentrations near the detection limit (0.3 nM) at 200 m (Beale, Dixon, Arnold, Liss, & Nightingale, 2013), while methanol and acetaldehyde concentrations at depth




did not decrease as rapidly. We note that for these measurements a flow of dry nitrogen was used as a background which may be an underestimation of the true system blank for acetaldehyde (see appendix). The chief advantage of using the bottom water measurement as the background is that the headspace gas has the same properties (humidity, temperature, exposure to the equilibrator) after equilibration as the headspace equilibrated with surface water and has been through the same collection protocol as the surface water samples.

The final blank we determined was the "wet equilibrator" blank.  This consisted of stopping the water flow into the equilibrator and purging the wet equilibrator (that had been coated with bottom seawater) with zero air for 20 min. During this blank measurement, humidity in the headspace remained constant as small water droplets remained inside of the coil and were not substantially dried by the zero air.  During the Arctic cruise, the wet equilibrator blank consistently resulted in the lowest blank reading on the PTR-MS for all VOCs except for methanol and acetone as a result of a contamination (discussed below). Thus, in practice the wet equilibrator blank seems to be the best surrogate for a "true" water blank for almost all VOCs measured here.

The 200 cm$^3$ jar used for separating the headspace from the seawater after equilibration is made from PTFE, and thus was not expected to cause any contamination. However, we found that the empty equilibrator blanks of methanol and acetone were about 0.2 ppb higher than their deep water blanks during the Arctic cruise.  The most plausible explanation for this seems to be an emission of methanol and acetone from the PTFE jar itself, which is suppressed during the water measurement. During the three-week field deployment, we observed a strong correlation between zero air and bottom water measurements ($R^2$= 0.92 methanol, $R^2$= 0.69 acetone), suggesting the concentrations of these VOCs at depth are either uniform or very small.  Because of the contamination as described above, we report seawater acetone and methanol concentrations from this cruise using bottom water as a blank; these concentrations should thus be viewed as lower limit estimates. After the cruise, we replaced the PTFE jar with a PTFE tee fitting (Swagelok) and this contamination greatly decreased.

We find that ultrapure MilliQ water or bottom seawater water is typically free of the less soluble compounds such as DMS, toluene, benzene and isoprene. This is confirmed by good agreements between the wet equilibrator blanks and the MilliQ/ bottom seawater measurements. The agreements also suggest that our system is not affecting the measured concentrations of these compounds e.g. through cell lysis. The concentrations of methanol, acetone, and acetaldehyde measured in the MilliQ water are much higher than those in seawater and are highly variable. Thus, we conclude that MilliQ water is not free of these gases (see appendix). Similarly, we found that tap water or bottled drinking water is typically not free of methanol, acetone, and acetaldehyde, likely due to slow leakage of these compounds from the piping or plastic container.

**3.2 Estimation of equilibration efficiency**

As a brief recap, for gases that appear to fully equilibrate in the SFCE, seawater concentrations ($C_w$) are computed from the equilibrator headspace mixing ratio ($C_a$) using the dimensionless Henry solubility constant (H) (R. Sander, 2015). Headspace equilibrator mixing ratios are converted from ppb to nmol dm$^{-3}$ using the ideal



gas law and the dilution of equilibrator headspace is accounted for by multiplying measured equilibrator mixing ratios by 1.2 (Sect. 3, Eq. (2)).

Where possible, values for Henry solubility recommended by S. P.Sander et al., (2015) were used for this calculation as those were deemed most reliable. These values represent freshwater solubilities and are converted to seawater solubilities accounting for salting out effects (Johnson, 2010). Values of the dimensionless Henry solubility (water over gas) in freshwater and seawater as well as the references for the solubility are displayed in  Table 1.

Two methods are used to assess the equilibration efficiency of the SFCE: evasion and invasion.  In evasion experiments, liquid standards of methanol, acetone and acetaldehyde were prepared by serial dilution of the pure solvent in the same batch of MilliQ water. Aliquots of pure, undiluted methanol (For spectroscopy Uvasol) and acetone (HPLC standard) were dispensed using volumetric pipettes. A 1 $cm^3$ volumetric flask was used to aliquot pure acetaldehyde (>=99.5%, A.C.S. Reagent). Subsequent dilutions utilised a volumetric pipette
and volumetric flask. Liquid standards of Isoprene and DMS were prepared gravimetrically airtight each day. A syringe pump (New Era Pump Systems) was used to dynamically dilute DMS and isoprene standards in a flow of MilliQ water. For this calibration, the flow of MilliQ water is measured at the drain.

For evasion experiments, a solubility-dependent fraction of dissolved VOCs is transferred into the gas phase during the equilibration process. Thus, the final dissolved concentration will be somewhat lower than the
initial concentrations. To account for the removal of a fraction of these gases from the waterside during equilibration, a purging factor (PF) based on mass conservation is applied. The PF is the ratio between the waterside concentration before and after complete equilibration in the coil. The derivation of this compound specific purging factor is presented in the appendix.  At equal air and water flow rates, it simplifies to:

$$PF = \frac{C_w(before\ equilibration)}{C_w(after\ equilibration)} = \frac{1}{H} + 1 \qquad (4)$$

For freshwater, computed purging factors assuming full equilibration and equal zero air: water flows are:
1.00015 for methanol, 1.00111 for acetone, 1.00225 for acetaldehyde, 1.063 for DMS, 1.184 for benzene, 1.209 for toluene and 2.568 for isoprene. The same computation in seawater gives the following purging factors: 1.00015 for methanol, 1.00122 for acetone, 1.0025 for acetaldehyde, 1.075 for DMS, 1.221 for benzene, 1.255 for toluene and 2.961 for isoprene. We see that PF varies from being insignificant (~=1) for highly soluble VOCs to quite large (>>1) for the sparingly soluble gases. To compute the expected headspace mixing ratio during
waterside calibrations (i.e. evasion experiments) assuming full equilibration, the known waterside concentrations are divided by the purging factor. For such evasion experiments, the equilibration efficiency is calculated as the measured headspace mixing ratio divided by the expected headspace mixing ratio assuming full equilibration.

We also tested the uptake of gaseous VOCs into the water phase (i.e. invasion). This is especially useful
for gases such as benzene and toluene, as we were unable to generate liquid standards of these compounds due to their toxicity. During invasion experiments, a flow of VOC gas standard was diluted to varying degrees with VOC-free zero air using mass flow controllers. This diluted VOC gas standard was then equilibrated with



essentially VOC-free Milli-Q water. This assumption is acceptable as we used relatively high carrier gas VOC mixing ratios (up to 50 ppb) and the Milli-Q water is essentially free of DMS, benzene, toluene and isoprene
(Sect. 3.1). The headspace equilibrator mixing ratio is measured and compared to the expected mixing ratio at full equilibration. Calculation of the expected mixing ratios at full equilibration during invasion experiments is presented in the appendix. For invasion experiments, the equilibration efficiency is calculated as the observed change in mixing ratio over the expected change in mixing ratio.

**4 System performance**

**4.1 Effect of humidity on the PTR-MS measurements**

        Equilibrator headspace mixing ratios were initially computed using compound specific rate constants of the reaction between the VOC in question and the hydronium ions in the drift tube of the mass spectrometer (Yang et al. 2013; Zhao and Zhang, 2004). Pre and Post cruise dynamic gas phase calibrations using a gravimetrically prepared VOC gas standard (Apel-Riemer Environmental Inc., Miami, Florida, USA, 517 ppb
acetaldehyde, 490 ppb methanol, 512 ppb acetone, 491 ppb isoprene, 527 ppb DMS, 500 ppb benzene, 483 ppb toluene) and two Bronkhorst mass flow controllers agree within 15% of the computed mixing ratios for all VOCs except isoprene. Isoprene was found to fragment significantly where 17% of the isoprene molecules are found at the parent ion (m/z 69) and 30% and 53% were found at m/z 41 and 39 respectively. This is in general agreement with Schwarz et al., (2009). This fragmentation ratio increases with increasing drift tube voltage (see
appendix).

        The PTR-MS measurements can be affected by humidity. To check for the effect of humidity on the PTR-MS measurement, gas calibrations were carried out at different humidities using three Bronkhorst mass flow controllers. To produce carrier gas air at different humidities, a flow of moist air saturated in humidity at 20°C is generated by passing dry clean air through a wetted SFCE and diluted by varying amounts of dry air from a gas
canister. The air is scrubbed by the Pt-catalyst which does not appear to change the humidity levels and added to the flow of VOC gas standard. The signal of most VOCs monitored is independent of the sample humidity. However, isoprene, benzene and toluene show a weak humidity dependence in their gas phase calibrations. Changing the humidity in sample air from completely dry to nearly saturated in humidity at 20°C, the abundance of isoprene parent ion increases by 33% (see appendix). This is because the hydronium water
clusters do not cause isoprene fragmentation upon ionisation (Schwarz et al. 2009). The opposite is observed with benzene and toluene where parent ion abundance was found to decrease by 12% and 18% respectively over same humidity range. This is because the humidity makes ionisation less efficient (see appendix). In fact, hydronium water clusters have a lower ionisation energy, thus ionising benzene and toluene less effectively (de Gouw and Warneke 2007; Warneke et al. 2001). The humidity dependant slopes from the gas phase calibrations
were used to correct the measured equilibrator headspace mixing ratios. We note that our use of a dilution flow lowers the humidity in the sample gas by 20% and thereby reduces the measurement sensitivity to humidity.





### 4.2 Measurement sensitivity toward air:water flow ratio

The air to water ratio of 100 $cm^3$n:100 $cm^3$ is chosen to allow for a reasonably long equilibration time, large surface area for exchange, and so high signal while satisfying the air flow requirements of the PTR-MS. They are also chosen such that the stripping of the soluble compounds from the water phase during equilibration would be small (i.e. purging factor near 1). Additionally, equal flows of air and water simplifies the calculation of waterside concentrations. The water flow was not monitored during the Arctic deployment and decreased by up to 20% due to aging of the peristaltic pump tubing. This could influence our measurement through at least (i)

the equilibration time and hence the efficiency in the coil; (ii) the purging factor. To investigate the influence of these competing factors on the signal, an experiment was performed after the cruise measuring the same solution of liquid standard at different water flows into the equilibrator where the air flow was kept constant (Fig. 2).

The signal of acetone, acetaldehyde, and DMS was found to be independent of the water flow into the

equilibrator. These results provide strong experimental evidence that i) VOCs with solubility that is greater than or similar to DMS equilibrate in the coil, and ii) the gas flow does not remove a large fraction of these gases from the water phase during the equilibration process (i.e. purging). In contrast, the signal of isoprene was found to decline with declining water flow. As the water flow is decreased in this experiment, the purging factor increased at a comparable rate as the isoprene headspace mixing ratios decreased.  This suggests that the

change in purging factor is largely responsible for the change in the isoprene signal (Fig. 2). Consequently, compared to the soluble VOCs there is an additional uncertainty of ~20% that is due to the variable water flow during the cruise (see Sect. 4.5).

### 4.3 Equilibration efficiency

Our aim is to build an equilibrator that fully equilibrates for the very soluble OVOCs. By maintaining a stable

equilibration efficiency of 100%, this design would maximise the signal to noise ratio and minimize the measurement uncertainty. This would also reduce the need for frequent calibrations. Robust waterside calibrations were not performed during the Arctic deployment due to logistical constraints. Post-cruise calibrations were carried out on an approximately weekly basis over several weeks, which is meant to be representative of the duration of the cruise. These calibrations were used to assess the equilibration efficiency

of SFCE and uncertainties therein.

Prior experiments with a similar setup suggest that the 10 m segmented flow tube presented here is at least a factor of two longer than required for full equilibration of DMS (Blomquist et al., 2017). Hence we expect the OVOCs (methanol, acetone and acetaldehyde) to fully equilibrate due to their higher solubility and airside control in gas exchange (Liss and Slater, 1974). Figure 3 shows calibration curves for DMS and isoprene using

liquid standards (i.e. evasion) over several weeks.  The calibration curve for DMS suggests full equilibration (Fig. 3a), where a ~5% underestimation of DMS in the mean is within the uncertainty of the solubility. The DMS calibration curves show very little noise and low weekly variability (±4% std. dev.), suggesting that the SFCE-PTR-MS setup is very stable. The calibration curve for isoprene suggests 62% equilibration efficiency (Fig. 3b). A



greater variability on a weekly basis (±14% std. dev.) is observed in the isoprene calibration curves, likely due to incomplete (and hence less consistent) equilibration.

Results from the invasion experiments are displayed in Fig. 4 and confirmed that the equilibrator fully equilibrates for DMS, as the measured and expected gas phase mixing ratios of DMS match. The equilibration efficiency of the less soluble gases benzene and toluene were found to be 94±1% and 95±2% respectively. The 5% difference is within the uncertainty of the solubility of these compounds, and so for computation of their

seawater concentrations we assume that these compounds fully equilibrate. This invasion experiment was also performed for the highly soluble OVOCs (methanol, acetone and acetaldehyde).  These gases were found to be entirely absorbed into the water phase, leading to essentially noise in the measurements of headspace mixing ratios.

The equilibration efficiency of isoprene (the least soluble compound that we measure by far) of 69% from

invasion is similar to what was measured in the evasion experiments (62%) if we use the isoprene solubility from Karl et al., (2003) and the temperature dependence from Leng et al., (2013).  We note that there is a large range in the isoprene solubility in the literature. Using the solubility values from Yaws & Yang (1992), Leng et al., (2013) or Mochalski et al., (2011) would result in a large discrepancy in the equilibration efficiency of isoprene between the evasion and invasion experiments, which we do not expect.

For the OVOCS, both theoretical considerations (e.g. Liss and Slater, 1974) and experiments with varying air:water flow ratio (Sect. 4.2) indicate that they should fully equilibrate within the SFCE. The average slope of 11 calibration curves for acetaldehyde and 14 calibration curves for methanol and acetone over a three-months period are shown in Fig. 5. Results are compared to the expected mixing ratio computed using every experimentally determined solubility listed in the compilation by R. Sander (2015). The measurements are also

compared to the solubility recommended by S. P. Sander et al. (2015) which was chosen as a critical synthesis of published solubilities.

The experimentally determined calibration slopes for OVOCs are very linear (typical $R^2$ above 0.95). However, in the mean they are about 1.5 times higher than expected compared to the solubility recommended by S. P. Sander et al. (2015). Nevertheless, these experimental mean slopes are within the range of published solubility values.

The relative standard deviation in the OVOC calibration curves (~25%) are much larger than that in the DMS calibration curves (4%), with the latter an indication for the stability of the PTR-MS/equilibrator system. On a weekly basis, the calibration curves of individual OVOCs correlate with each other, and these OVOCs were diluted together from pure reagents. This suggests that most of the observed variability in OVOC calibration from week to week is due to errors or contamination in the serial dilution procedure. In order to ensure consistency with

previous equilibrator setups (Kameyama et al., 2010), we report our waterside concentrations using recommended solubilities from S. P. Sander et al. (2015).  Using the mean of the experimental waterside calibrations would decrease the computed OVOC seawater concentrations by approximately 50%.

**4.4 Measurement response time**

To determine the response and delay time of the equilibrator and to test for a possible memory effect due to

wall adsorption and desorption effects, MilliQ water was measured followed by a solution of liquid standards



containing 20 nmol dm$^{-3}$ acetone, 20 nmol dm$^{-3}$ acetaldehyde and 200 nmol dm$^{-3}$ methanol (Fig. 6). Discrete samples were swapped over rapidly and the water flow into the equilibrator was stopped briefly to avoid interfering with the measurement.

The residence time (0.6 min) in the equilibrator segmented flow tube was calculated from the flow of air and water into the equilibrator and the volume of the segmented flow tube. The response time (e-folding time) of the equilibrator response to the step change was calculated using the 8s PTR-MS measurements. It was calculated to be 35, 33 and 33 s for methanol, acetone and acetaldehyde respectively. Thus, the response time appears to be independent to the solubility of the compound. The sharpness in the rate of increase/decrease upon water change in this experiment also suggests that there is no obvious 'carry over' or memory effect.

While the response time of the SFCE is less than 1 min, to reduce random noise and improve the precision of the measurement, we typically average measured equilibrator headspace mixing ratios over 6 min for both underway measurements and the discrete measurements.

### 4.5 Measurement precision and Limit of detection

The measurement precision and the limit of detection (LOD) of this system are dictated by the noise of the PTR-

MS measurement. This in turn depends on the dwell time of the detector at a given mass and thus the time the data are averaged over. Additionally, it depends on the factors stated in Eq. (2), namely the solubility of the compound, the purging factor and the dilution flow. For isoprene, the measurement precision depends on the factors stated in Eq. (3), where an additional consideration is the variability in water flow, which adds 20% uncertainty. We compute the measurement precision as the standard deviation (1 $\sigma$) of 10 consecutive 6 min wet

equilibrator blank measurements, which is then converted to a waterside concentration using Eq. (2) and Eq. (3) for isoprene. This results in averaging the noise over 70 measurement cycles with a dwell time at each mass of 500 ms giving an effective dwell time of 3.5 s. The LOD is defined as 3 $\sigma$. The resulting measurement noise and limit of detection for each compound are displayed in table 2 for 6 min averaged data. These values should approximately halve if the data are averaged to 30 min intervals instead.

In the case of discrete samples, a larger water volume should improve the measurement precision by allowing for a longer sampling and averaging time. This is especially relevant for the most soluble compounds including methanol or acetone.

### 5 Field measurements

The SFCE coupled to PTR-MS was used to measure underway surface water and depth profiles in the Canadian

Arctic on board the Ice Breaker CCGS *Amundsen* from mid-July until the beginning of August 2017. The ship went from Iqaluit to Smith Sound and ended near Resolute (cruise track map in appendix). A selection of data is displayed in Fig. 7, but the full dataset will be presented elsewhere. Surface underway measurements were made using the ship's main built-in underway water supply (3–4 m depth, inlet located at front starboard side of the ship), whereas the discrete CTD measurements were from Niskin bottles at 5 m depth.



The SFCE coupled PTR-MS allows for continuous measurement of a breadth of VOCs at a high resolution. Sample data presented in Fig. 7, contains 5 min measurements that are further averaged to hourly intervals. The underway surface water measurements capture a larger variability of concentrations (e.g. acetone 3.5–23 nmol dm$^{-3}$) than discrete surface samples collected from the ship's rosette (e.g. acetone 2.9–10 nmol dm$^{-3}$). This highlights one of the benefits of underway measurements, as some of these compounds display noticeable fine

scale variability likely due to their short lifetime.

Contamination of the underway water supply relative to the CTD Niskin bottle has been observed for acetone (Yang et al. 2014), probably due to the ubiquity of OVOCs and their wide application in shipboard science (e.g. acetone for Chlorophyll extraction). Previous underway measurements of isoprene and DMS have found that after switching the underway water supply on, the first few hours of data typically showed significantly higher

concentrations (Andrews et al. 2015). For our dataset on this Arctic cruise, underway measurements and discrete samples from 5 m depth do not show any obvious discernible difference for most of the VOCs. This is confirmed by the fact that the average concentration reported from the 5 m Niskin bottle (+/- 95% confidence interval of the mean) overlaps with the average concentration measured from the ship's built-in underway water inlet 3h either side of the CTD measurement (Table 3). Measurements below the limit of detection were

included for all analysis to avoid a biased mean. The DMS and toluene concentrations from the underway water inlet appear to be higher compared to measurements from the 5 m Niskin bottle. This could possibly be due to a contamination in the underway water supply or due to the different sample depths between the underway and CTD data. We observed strong vertical gradients near the surface for most of the VOCs, which will be discussed in more detail in a future manuscript. The data presented here have not been corrected for this

possible contamination.

**6 Conclusion and recommendations**

This paper presents a ship-based equilibrator system coupled to a PTR-MS for measurements of a breadth of gases in seawater. Its main advantage lies in its unique design. The segmented flow gives a high degree of equilibration due to surface renewal within each water segment (Xie et al., 2001), a large surface area for gas

exchange, and a long equilibration time. We find that it fully equilibrates for gases of similar or higher solubility than toluene. The unique air-water separation system allows for rapid drainage of water without droplet formation, thus yielding a high response time of less than 1 min even for the highly soluble OVOCs. Additionally, the SFCE can be used for underway and discrete sampling due to the ease of changing the water intake and low water flow requirements (100 cm$^3$ min$^{-1}$). Since it consists entirely out of commercially available PTFE tubing, it

can be easily and relatively cheaply constructed and should have minimal wall adsorption effects. The smooth surfaces and constant water flow make the equilibrator easy to clean and fairly resistant to biofouling. Finally, the SFCE system is designed with multiple fail-safes, such that a failure of an individual component does not cause the equilibrator to floor or over/under-pressurise.

The equilibrator can be used to measure compounds that only partially equilibrate (e.g. isoprene) but with

slightly higher uncertainty than for fully-equilibrating compounds. The SCFE could easily be optimised for measuring these less soluble gases by making the segmented flow tube longer to allow more time for



equilibration or by adding an isotopically labelled standard. One of the considerations when measuring dissolved gases with PTR-MS is the effect humidity on the signal. We have presented discussions on how to estimate the background of the water measurement and how to correct for the effect of humidity on the signal
of the PTR-MS (see appendix). Further work is being carried out to obtain a more robust background estimate that does not require deep seawater samples.

The SFCE-PTR-MS was used to measure methanol, acetone, acetaldehyde, DMS, isoprene, benzene and toluene on board the Canadian ice breaker CCGE *Amundsen* during Arctic spring. A selection of the underway measurements is presented here with a comparison to samples obtained from 5m Niskin bottles.

We envisage wide applications of this novel equilibrator such as deployment on further research cruises for measurement of a breadth of gases. The SFCE can be coupled to other gas phase detectors such as a CIMS (Saltzman et al., 2009). The equilibrator can also be incorporated into existing methods that require fast response times, for example near-surface ocean profilers (Sims et al., 2017). The SFCE can also be used to perform underway photochemical or surface microlayer reaction experiments. Finally, the method can be used
to improve the measurements of dissolved gases in algae cultures, which currently rely on discrete headspace sampling (Halsey et al., 2017).

**7 Appendix:**

**Appendix A: PTR-MS Settings**

To measure the VOC concentrations, we use a commercially available high sensitivity Proton-Transfer-Reaction
Mass Spectrometer (de Gouw & Warneke, 2007; Lindinger & Jordan, 1998). Briefly, water vapor is ionised in a hollow cathode. The hydronium ions react with sample air in the drift tube. Here, gases with a proton affinity higher than water, including many VOCs, are ionised continuously usually without fragmentation. Hydronium ions are in large excess of the VOCs, which allows for application of pseudo-first order kinetics in the drift tube. Together with relatively well-studied reaction rates between VOCs and hydronium ions (Zhao & Zhang, 2004)
and mass spectrometer specific parameters (Yang et al. 2013), the mixing ratios of the VOCs can be fairly accurately computed without the need of an internal standard (Lindinger & Jordan, 1998). Nevertheless, reaction rate constants between VOCs and hydronium ions have a reported error margin of up to 50% (Blake et al. 2009; Ellis and Mayhew 2014). To correct for this, dynamic gas phase calibrations were carried out using a certified gas standard.

SFCE headspace is laden with humidity, which influences the PTR-MS measurement. Previous observations suggest that humidity in the sample affects drift tube kinetics through the formation of hydronium water clusters. In practice, such water dimmers are monitored at m/z 37 (i.e. isotopic hydronium water cluster $(H_2^{18}O^+)H_2O$) as a percentage of the primary ion count, accounting for isotopic abundance (Blake et al. 2009):

$$H_2O + H_3O^+ \rightarrow H_3O^+(H_2O) . \qquad (A1)$$





Humidity has several effects on the measurement: (i) The additional water molecule stabilises the primary ion
       by sharing the positive charge thus increasing its proton affinity (Blake et al. 2009). For example benzene and
       toluene possess intermediate proton affinities and are ionised by the primary ion, but not the water cluster
       (Warneke et al. 2001). On the other hand, ionisation by the water cluster is softer and hence leads to less
       fragmentation for example of isoprene in the drift tube (Schwarz et al. 2009); (ii) humidity in the drift tube leads

to non-collision rate limited reactions; (iii) large water clusters have different ion mobility, where (ii) and (iii) are
       not accounted for in the computation of the measured mixing ratio (Blake et al. 2009); (iv) sample humidity
       affects the backgrounds of some of the VOCs monitored (de Gouw & Warneke, 2007). (v) Some PTR-MS have a
       collision-induced dissociation (CID) chamber at the end of the drift tube in which the E/N is briefly raised to
       simplify the mass spectra and remove humidity induced clusters which leads to an overestimation of the true

hydronium primary ion concentration in the drift tube and thus an overestimation of VOC concentrations (Blake
       et al. 2009). However, our PTR-MS instrument does not have a CID chamber. (vi) It is tempting to think of VOC
       protonation as a one way reaction. However the reaction is reversible and an increase in the water vapor
       concentration leads to increased reverse reaction which is strongly temperature dependant (Blake et al. 2009).

       While the effect of changing humidity on the PTR-MS signal could in theory be corrected for based on
complex parametrisations (Kameyama et al., 2010), keeping the sample humidity constant greatly simplifies the
       corrections. Partly because of this, we maintain a constant humidity in SFCE headspace (monitored at m/z of 37)
       by keeping the coil at 20°C. A dryer (e.g. Nafion) has been successfully used in the measurements of seawater
       DMS (Blomquist, Huebert, Fairall, & Faloona, 2010) and would reduce many of the aforementioned
       measurement uncertainties.  However Nafion dryers are known to remove very soluble/reactive OVOCs

(Kameyama et al., 2010) and thus is not an option for our measurements.

       Clearly, excessive water clustering in the drift tube is undesirable. To keep the water dimer to be < 5%
       of the primary ion count when measuring headspace equilibrator, the PTR-MS drift tube was operated at 160Td
       (700V, 2.2 mBar and 80°C in the drift tube). The water vapor flow into the source was set to 5 $cm^3n$/min, the
       source current at 3 mA and the source valve to 35%. At these settings, the amount of hydronium water clusters

is below 1% when measuring dry zero air and the amount of $O_2^+$ ions is below 0.7% of the primary ion counts.
       Residual water cluster measured during dry canister measurement is due to unionised water vapor from the
       hollow cathode entering the drift tube (Warneke et al. 2001).

       The PTR-MS is deployed in selective ion mode. Ions monitored at m/z 33, 45, 59, 63, 69, 79 and 93 were
       attributed to methanol, acetaldehyde, acetone, dimethyl sulphide, isoprene, benzene and toluene in
accordance with previous mass assignments (Williams et al. 2001; Warneke et al. 2003). Propanal has previously
       been shown to have a very minor contribution to m/z 59 (Beale et al., 2013). For methanol, we correct for the
       oxygen isotope interference by monitoring $O_2^+$ in the drift tube and applying a theoretical isotopic distribution
       ratio, which is 0.076% of the $O_2^+$ signal.

**Appendix B: Humidity experiments and fragmentation experiments**

To investigate the effect of humidity on the background, we measured clean synthetic air at different
       humidities. VOC-free air saturated in humidity is generated by passing synthetic air (BTCA grade) over a wetted





SFCE coated inside with MilliQ water and kept at 20°C in a water bath. The concentration of water vapor was calculated to be 22.9 millimole water vapor per mole of air. The water drain was capped for this experiment to balance out the pressure resistance provided by the Pt-Catalyst. This air is scrubbed with a Pt-Catalyst to oxidize
all VOCs to $CO_2$. Efficiency of this catalyst at oxidizing VOCs in wet and dry air is demonstrated elsewhere (Yang and Fleming 2019) and it was found that the catalyst did not affect humidity levels. This flow of air saturated in water at 20°C is dynamically diluted with dry zero air to generate VOC-free air at different humidities.

To verify that the measurement is not excessively affected by humidity at those settings, dynamic gas phase calibration curves were carried out at different humidity levels. Dynamic gas phase calibrations were
carried out using mass flow controllers to dilute a flow of zero-VOC air at different moistures (BTCA air scrubbed by Pt-catalyst) and a gravimetrically prepared standard gas in ultrapure $N_2$ with known amount of VOC (517 ppb acetaldehyde, 490 ppb methanol, 512 ppb acetone, 491 ppb isoprene, 527 ppb DMS, 500 ppb benzene, 483 ppb toluene, Apel–Riemer Environmental Inc., Miami, Florida, USA). The ratio between synthetic air and VOC standard in $N_2$ was typically more than 10:1, thus not significantly changing the matrix. During these
experiments and during the Arctic field deployment, the abundance of water dimer was monitored at m/z 37 (representing $(H_2{}^{18}O^+)H_2O$) and calculated as a fraction of the primary ion while accounting for its specific transmission efficiency. Primary ion was monitored as the water isotope at m/z 21 and multiplied by the isotopic ratio of 500;

$$\%m/z\ 37 = \frac{cps(m/z\ 37)}{cps(m/z\ 21) * 500} \tag{B1}$$

Measurement of zero air at different humidities showed an exponential dependence of the DMS (m/z 63) and acetaldehyde (m/z 45) background to the humidity of the sample air (Fig. B1). The background of the other compounds presented here remained unaffected by humidity. For this analysis, dry zero air was subtracted as a system blank thus the Fig. B1 represents the additional background due to sample humidity. Here, a measured m/z 37 of 1% corresponds to dry canister air measurement. A %m/z 37 between 1.4 and 2.0%
corresponds to outside air measurements and %m/z 37 of 2.2% corresponds to measurements of equilibrator headspace. The equilibrator headspace is nearly saturated in humidity as it is diluted by the dilution flow to reduce humidity.

Water vapor concentration and %m/z37 correlate linearly, thus both variables can be plotted on the same axis for comparison (Fig. B1). Lines of best fit for acetaldehyde and DMS background (in ppb) as a function
of the additional hydronium water cluster in the drift tube due to sample humidity (in % me37 of me21) was found to be;

$$ppb(DMS) = 0.351 - 0.789\ \{\%m/z\ 37\} + 0.425\ \{\%m/z\ 37\}^2 \tag{B2}$$

$$ppb(acetaldehyde) = 0.0128 - 0.257\ \{\%m/z\ 37\} + 0.224\ \{\%m/z\ 37\}^2 \tag{B3}$$





The flow of water vapor into the source was also varied while measuring clean dry canister air to simulate the influence of humidity induced water clusters on the background of the measurement. This indicated that the
background of all the masses monitored changed and significantly increased for the soluble OVOCs (methanol, acetone, acetaldehyde). This is suggesting that the high background in the measurement of these compounds is due to residues of OVOCs in the water reservoir filled up with ultrapure water by the manufacturer. Thus highlighting the difficulty of obtaining OVOC-free water.

The dynamic gas phase calibrations at different humidities showed that only calibration slopes of benzene, toluene and isoprene were humidity dependant (Fig. B2–Fig. B4). At the settings used, the calibration slopes of the other VOCs monitored did not vary with humidity. Benzene and toluene (Fig. B2 and Fig. B3) display humidity dependant calibration slopes because they possess intermediate proton affinities and are ionized by the primary ion but not by the primary ion water cluster (Warneke et al. 2001). The primary ion water cluster is more stable, because the additional water cluster stabilizes the positive charge (Blake et al.
610 2009).

For isoprene, the opposite effect is observed (Fig. B4) where the stabilising water cluster from humidity makes ionisation softer thus increasing the yield of the parent ion at m/z 69. Note that humidity dependant fragmentation of isoprene in PTR-MS has been observed before (Schwarz et al. 2009). Other masses that isoprene fragments can be found are m/z 39 and m/z 41.

To further investigate the fragmenting behaviour of isoprene, the same known amount of gas standard was measured at different voltages in the drift tube and ions at mass 41 and 69 were monitored (Fig. B5).

At lower voltages, the abundance of m/z 69 increases thus further supporting that isoprene is fragmenting in the drift tube. This fragmentation ratio was found to be very stable and vary by less than 5% over one month for twice weekly calibrations. The remaining isoprene molecules probably reside at m/z 39,
which was found to be the dominant ion in this fragmentation (Schwarz et al. 2009). For the isoprene measurements presented here, the fragmentation ratio of isoprene at those settings is accounted for.

**Appendix C: Map of the cruise track of the selection of data presented here**

A map of the cruise track of the underway data presented here is shown in Fig. C1.

**Appendix D: Derivation of the purging factor**

As mentioned in the main paper in Sect. 3.2, using equilibrator headspace mixing ratios, we compute waterside concentrations after equilibration in the coil. However, a solubility-dependent fraction of dissolved VOCs is transferred into the gas phase during the equilibration process. Thus the final dissolved concentration will be somewhat lower than the initial concentrations. To account for the removal of a fraction of these gases from the waterside during equilibration a purging factor (PF) based on mass conservation is applied. The PF is
the ratio between the waterside concentration before and after complete equilibration in the coil:





$$PF = \frac{C_w(before\ equilibration)}{C_w(after\ equilibration)} \quad\quad (D1)$$

, where;

$$C_W(before\ equilibration) = \frac{X_{tot}}{V_W} \quad\quad (D2)$$

Here $X_{tot}$ is the total number of moles in the system and $V_W$ is the volume of water. In the following demonstration, $X_{a_{fin}}$ and $X_{w_{fin}}$ are airside and waterside number of moles after equilibration and $V_a$ is the volume of carrier gas. Thus $X_{a_{fin}}$ is the number of moles measured as equilibrator headspace mixing ratios.

Hence;

$$C_W(after\ equilibration) = H * C_a(after\ equilibration)$$

$$\frac{X_{w_{fin}}}{V_w} = H * \frac{X_{a_{fin}}}{V_a}$$

$$X_{w_{fin}} = H * X_{a_{fin}} * \frac{V_w}{V_a} \quad\quad (D3)$$

Combining $X_{a_{fin}} = \frac{X_{w_{fin}}}{H*\frac{V_w}{V_a}}$ and $X_{tot} = X_{w_{fin}} + X_{a_{fin}}$ (D4) gives;

$$X_{tot} = X_{w_{fin}} + \frac{X_{w_{fin}}}{H*\frac{V_w}{V_a}}$$

$$X_{tot} = \left(1 + \frac{1}{H*\frac{V_w}{V_a}}\right) * X_{w_{fin}}$$

$$X_{w_{fin}} = \frac{X_{tot}}{1 + \frac{1}{H*\frac{V_w}{V_a}}} \quad\quad (D5)$$

Thus, $C_W(after\ equilibration) = \frac{X_{w_{fin}}}{V_w}$ (D6)

Combining Eq. (D5) and (D6) gives $C_W(after\ equilibration) = \frac{X_{tot}}{1 + \frac{1}{H*\frac{V_w}{V_a}}} * \frac{1}{V_w}$ (D7)

Combining Eq. (D2) and (D7) with Eq. (D1) gives:

$$PF = \frac{\frac{X_{tot}}{V_w}}{\frac{X_{tot}}{1 + \frac{1}{H*\frac{V_w}{V_a}}} * \frac{1}{V_w}}$$





$$PF = \frac{X_{tot}}{V_w} * \frac{1 + \frac{1}{H * \frac{V_w}{V_a}} * V_w}{X_{tot}}$$


$$PF = 1 + \frac{1}{H * \frac{V_w}{V_a}} \qquad (D8)$$

At equal zero air/water flow rates, this is simplified to:

$$PF = \frac{C_w(before\ equilibration)}{C_w(after\ equilibration)} = \frac{1}{H} + 1 \qquad (D9)$$

The expected mixing ratios during invasion experiments is calculated by combining Eq. (D4) and (D5);

$$X_{tot} = X_{a_{fin}} + X_{a_{fin}} * \frac{1}{H} * \frac{V_w}{V_a}$$

$$X_{a_{fin}} = \frac{X_{tot}}{1 + H * \frac{V_w}{V_a}} \qquad (D10)$$

, where the total number of moles is the diluted VOC gas standard mixing ratio.

**Appendix E: Compilation of experimentally determined solubilities for methanol, acetone and acetaldehyde**

**Appendix F: Photographs of the instrument setup**

**8 Author contribution**

CW and MY designed the equilibrator. CW carried out system performance tests. PN, AJ, DC and WS provided
input to the method development. CW carried out the deployment on board with help from MY during
installation. Collaboration with BE made these measurements in the Canadian Arctic possible. CW prepared the
manuscript with contributions from all co-authors.

**9 Competing interest**

The authors declare that they have no conflict of interest.

**10 Acknowledgements**

This work was supported by the Natural Environment Research Council through the EnvEast Doctoral Training
Partnership [grant number NE/L002582/1] and the United Kingdom & Canada Arctic Partnership: 2017 Bursaries
Programme funded by the UK Department for Business, Energy and Industrial Strategy. Thanks a lot to R. Beale
for her help during the early stages of the method development. Many thanks to M. Ahmed and B. Butterworth



(University of Calgary) for their invaluable help during the field deployment. Many thanks to D. Collins
(University of Toronto) for shipping the chemicals to port and J. Abbat (University of Toronto) for providing a
gas calibration standard. Thanks to the crew and the chief scientists on board the CCGS *Amundsen*, namely
Martine Lizotte and Jean-Éric Tremblay for accommodating this work.

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

**Table 1: Dimensionless Henry solubility values (water over gas) in freshwater and seawater used to compute dissolved concentrations.**

|  | Henry solubility at 20°C in freshwater | reference | Henry solubility at 20°C in seawater |
|---|---|---|---|
| methanol | 6716 | S. P. Sander et al., (2015) | 6494 |
| acetone | 901 | S. P. Sander et al., (2015) | 819 |
| acetaldehyde | 444 | S. P. Sander et al., (2015) | 400 |
| DMS | 15.78 | S. P. Sander et al., (2015) | 13.28 |
| benzene | 5.44 | Leighton and Calo, (1981) | 4.52 |
| toluene | 4.77 | McCarty and Reinhard, (1980) | 3.92 |
| isoprene | 0.638 | solubility from Karl et al., (2003) using temperature dependence from Leng et al., (2013) | 0.510 |

**Table 2: Precision and limit of the detection of the seawater VOC measurements (6 min average).**

|  | measurement precision $1\sigma$ | Limit of detection |
|---|---|---|
| methanol (nmol dm$^{-3}$) | 6.52 | 19.56 |
| acetaldehyde (nmol dm$^{-3}$) | 0.17 | 0.51 |
| acetone (nmol dm$^{-3}$) | 0.44 | 1.32 |
| DMS (nmol dm$^{-3}$) | 0.0069 | 0.0207 |
| isoprene (pmol dm$^{-3}$) | 0.58 | 1.74 |
| benzene (nmol dm$^{-3}$) | 0.0043 | 0.0129 |
| toluene (nmol dm$^{-3}$) | 0.0042 | 0.0126 |



**Table 3: Average concentration measured for each compound from the 5m Niskin bottle and 3h either side of the Niskin measurement**
**from the ship's build-in underway water inlet. Errors represent 95% confidence interval of this average.**

|  | 5m Niskin | Underway water inlet |
| --- | --- | --- |
| methanol (nmol dm$^{-3}$) | 17±6 | 15±6 |
| acetone (nmol dm$^{-3}$) | 7±2 | 8±2 |
| acetaldehyde (nmol dm$^{-3}$) | 3.8±1.2 | 3.8±1.0 |
| DMS (nmol dm$^{-3}$) | 0.90±0.16 | 1.51±0.38 |
| isoprene (pmol dm$^{-3}$) | 9.96±1.25 | 9.42±2.36 |
| benzene (nmol dm$^{-3}$) | 0.050±0.008 | 0.059±0.021 |
| toluene (nmol dm$^{-3}$) | 0.037±0.006 | 0.065±0.011 |

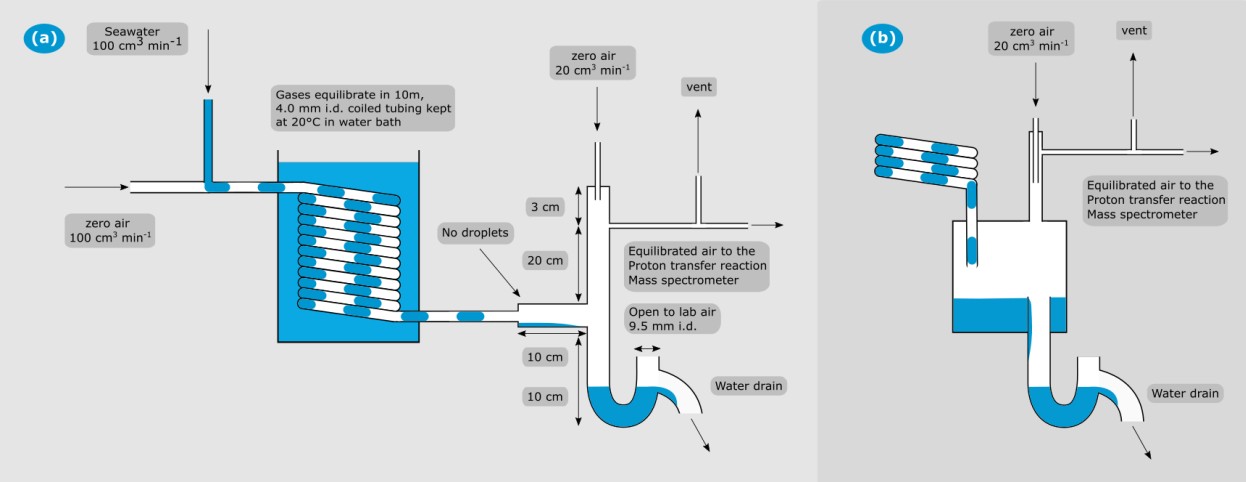

**Figure 1: A. Schematic of the segmented flow coil equilibrator coupled to PTR-MS. B. Schematic of the jar that was used during the Arctic deployment for headspace-water separation. All other connections were kept the same.**





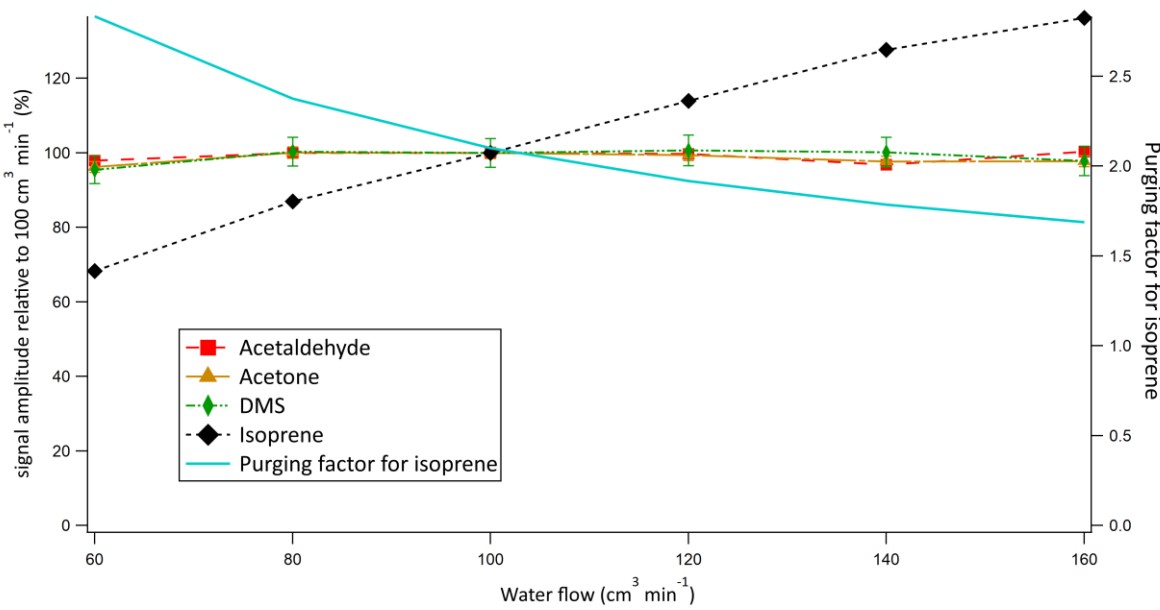

**Figure 2**: **Relative signal as a function of water flow into the equilibrator. Error bars represent random error propagation where the initial error has been determined from the standard deviation of 10 consecutive 6 min blank measurements.**







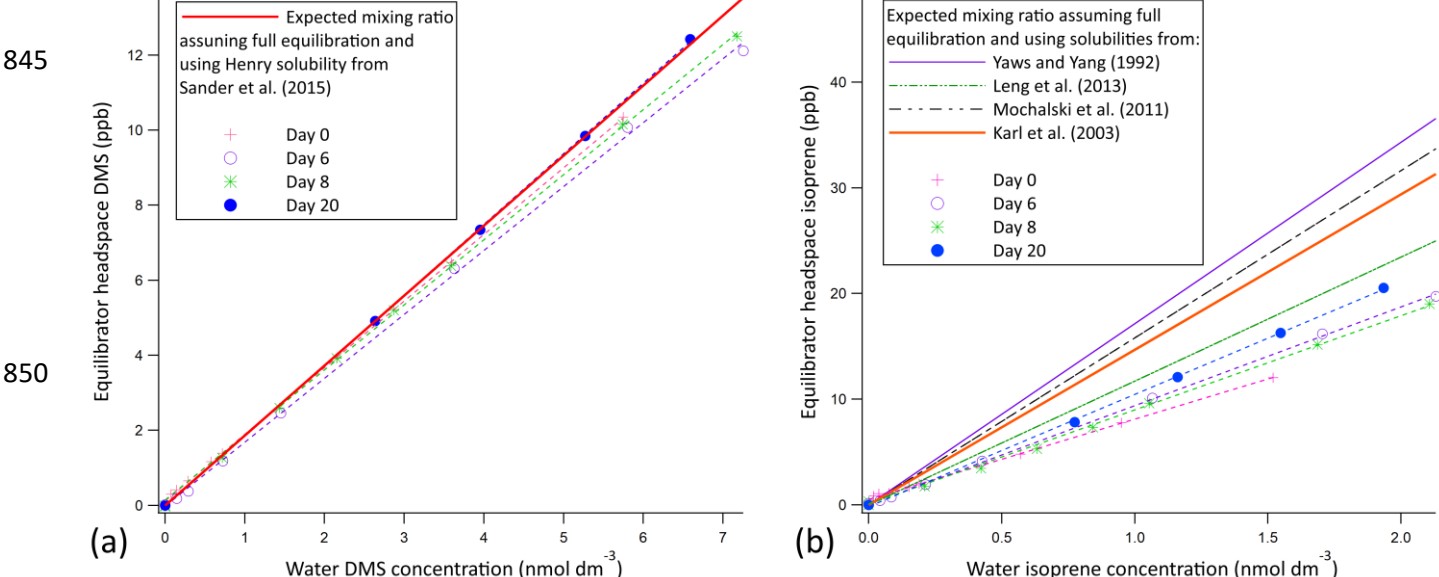

**Figure 3: Waterside calibration curves for DMS (panel a) and isoprene (panel b). Average slope of the experimental calibration curve was found to be 1.77 ppb nmol$^{-1}$ dm$^3$ ±4% and 9.12 ppb nmol$^{-1}$ dm$^3$ ±14% for DMS and isoprene respectively where errors represent standard deviation over a three-week period. Full equilibration slope was computed to be 1.87 ppb nmol$^{-1}$ dm$^3$ and 14.69 ppb nmol$^{-1}$ dm$^3$ for DMS and isoprene respectively (using S. P. Sander et al., (2015) for DMS solubility and Karl et al., (2003) solubility with Leng et al., (2013) temperature dependence). This suggests approximately 100% and 62% equilibration efficiency for DMS and isoprene respectively. Error bars are too small to display, but the noise associated with the measurement was found to be 0.0084 and 0.0044 ppb for DMS and Isoprene, respectively. This was calculated as the std. dev. of 10 consecutive water blank measurements.**







**Figure 4:** Invasion calibration curves for benzene (a), toluene (b), DMS (c) and isoprene (d) where a known amount of standard gas is added to the zero air carrier gas while measuring VOC-free Milli-Q water. Error bars were too small to display, but the noise associated with the measurement was found to be 0.0084 and 0.0044 ppb for DMS and Isoprene respectively and 0.015 and 0.013 ppb for benzene and toluene respectively. This was calculated as the std. dev. of 10 consecutive water blank measurements. A 1:1 line is included in 4 to illustrate the role of the water phase in absorbing these compounds.








**Figure 5:** Displayed are the average experimentally determined slopes of 14 calibration curves of methanol (a) and acetone (b) and 11 calibration curves of acetaldehyde (c). Shaded area indicates one sigma standard deviation of the variance in the slope during this three-month period. Average experimentally determined calibration slope for methanol, acetone and acetaldehyde were 0.00786 ±0.00115 ppb nmol⁻¹ dm³, 0.0469 ±0.0145 ppb nmol⁻¹ dm³ and 0.0743 ±0.0190 ppb nmol⁻¹ dm³. Plotted along this are the predicted slopes using



all experimentally determined solubilities as listed in R. Sander (2015). The recommended solubility by S. P. Sander et al., (2015) is plotted as a solid thick line in dark blue. The key to the figure is listed in the appendix, listing an in-text reference followed by the dimensionless water over air Henry solubility and the predicted slope using the experimentally determined solubility. For full reference of the cited solubilities, please refer to R. Sander (2015).

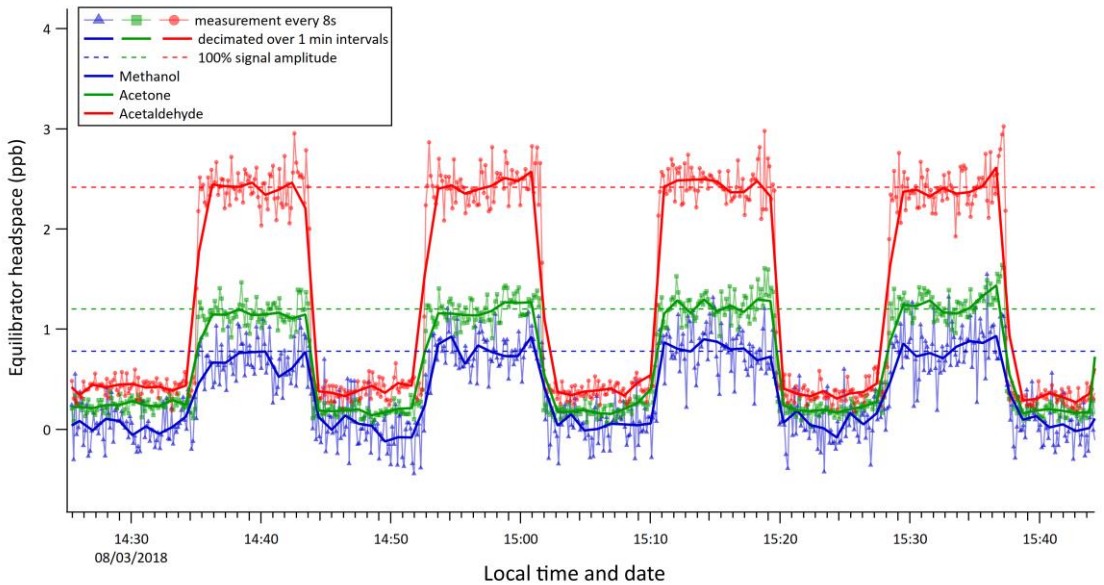

Figure 6: Instrument response to step changes in seawater OVOC concentration (step size: 20 nmol dm-3 acetone, 20 nmol dm-3 acetaldehyde, 200 nmol dm-3 methanol).





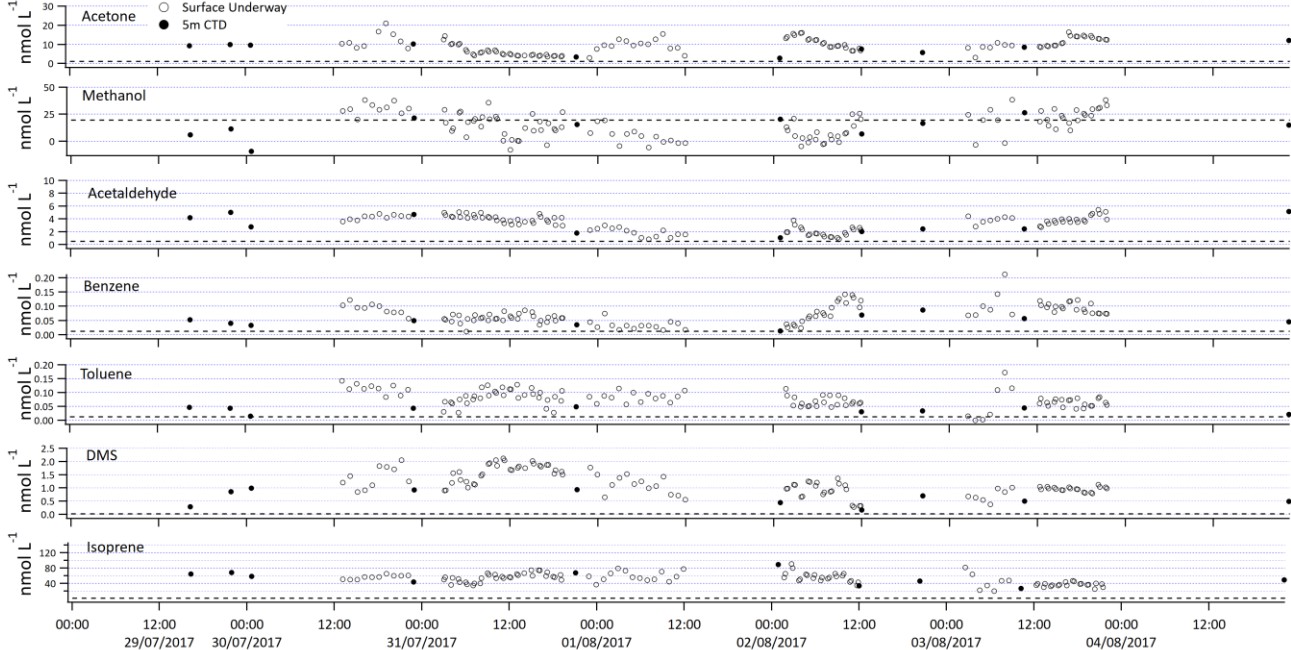


**Figure 7:** Selection of VOC measurements made from the ship's build in underway surface water supply (open symbols) and discrete samples from 5m rosette (closed symbols). The dotted line represents the limit of detection.

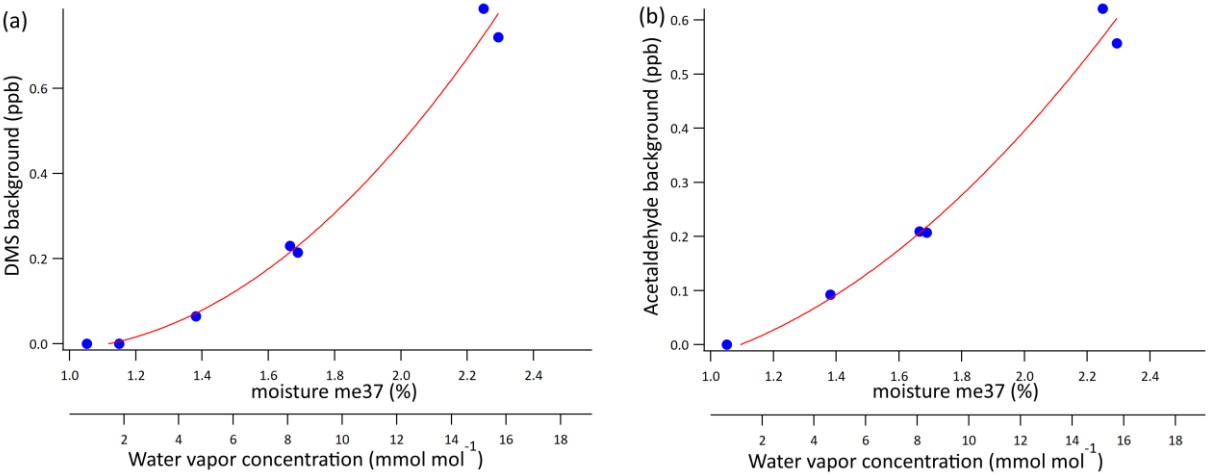

**Figure B1:** Background dependence of DMS (a) and acetaldehyde (b) signal on the humidity in the sample air. Error bars represent the standard deviation of ten consecutive blanks.




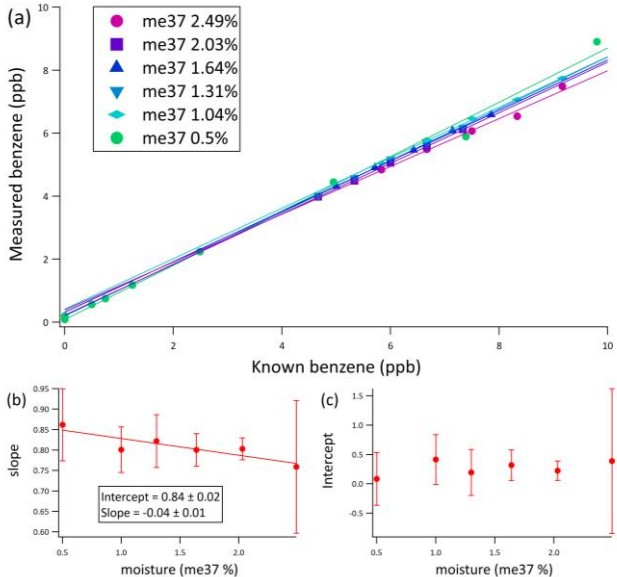

**Figure B2: (a) Benzene gas phase calibrations at different humidities and the dependency of the slope (b) and background (c) on the humidity. Error bars on the slope and intercept represent 95% confidence intervals of the linear regression.**

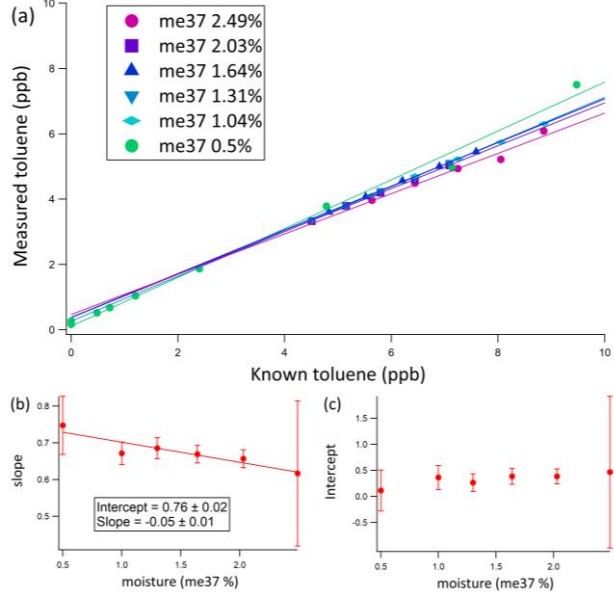


**Figure B3: (a) Toluene gas phase calibrations at different humidities and the dependency of the slope (b) and background (c) on the humidity. Error bars on the slope and intercept represent 95% confidence intervals of the linear regression.**



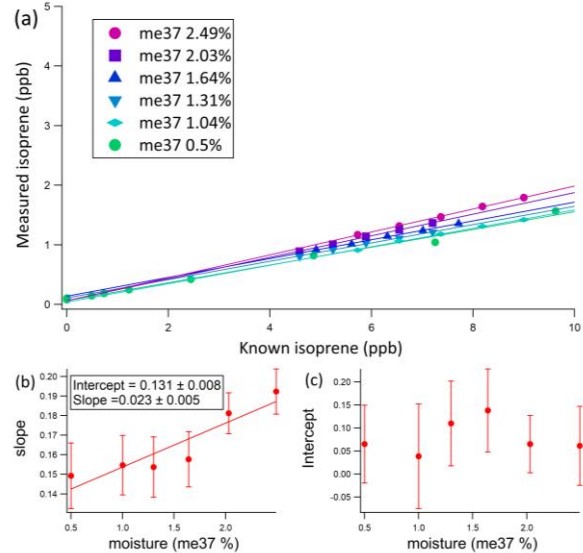

**Figure B4: (a) Isoprene gas phase calibrations at different humidities and the dependency of the slope (b) and background (c) on the**
**humidity. Error bars represent 95% confidence intervals of the linear regression.**

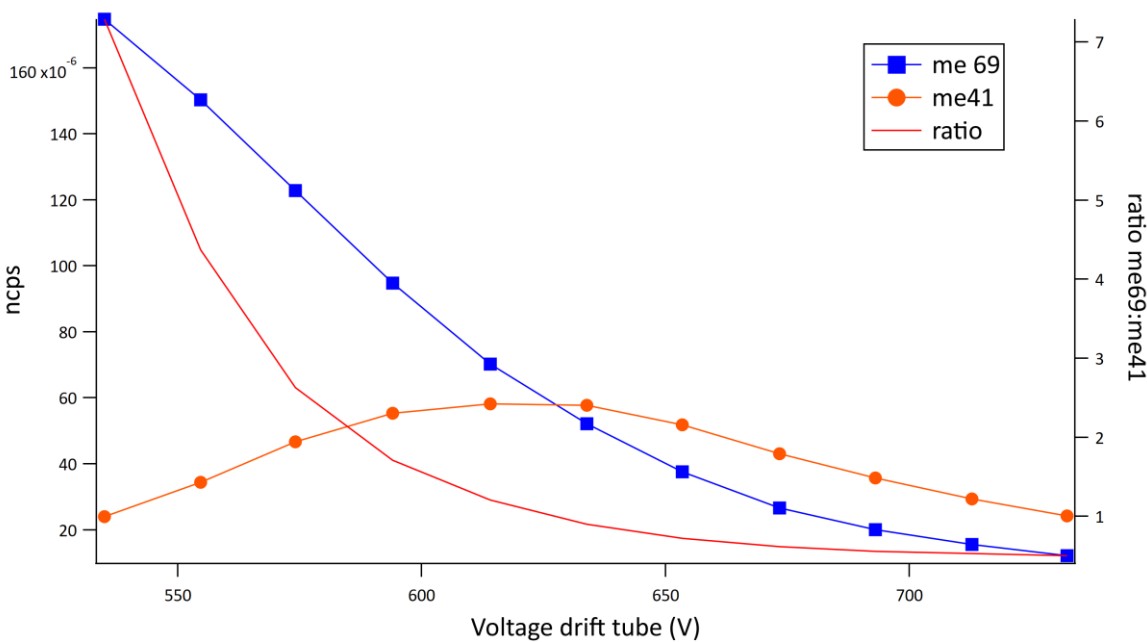

**Figure B5: Ratio of the isoprene parent ion at m/z 69 to the fragment at m/z 41 as a function of drift tube voltage. Plotted along are the measured isoprene mixing ratio computed from the mass of each of the ions.**





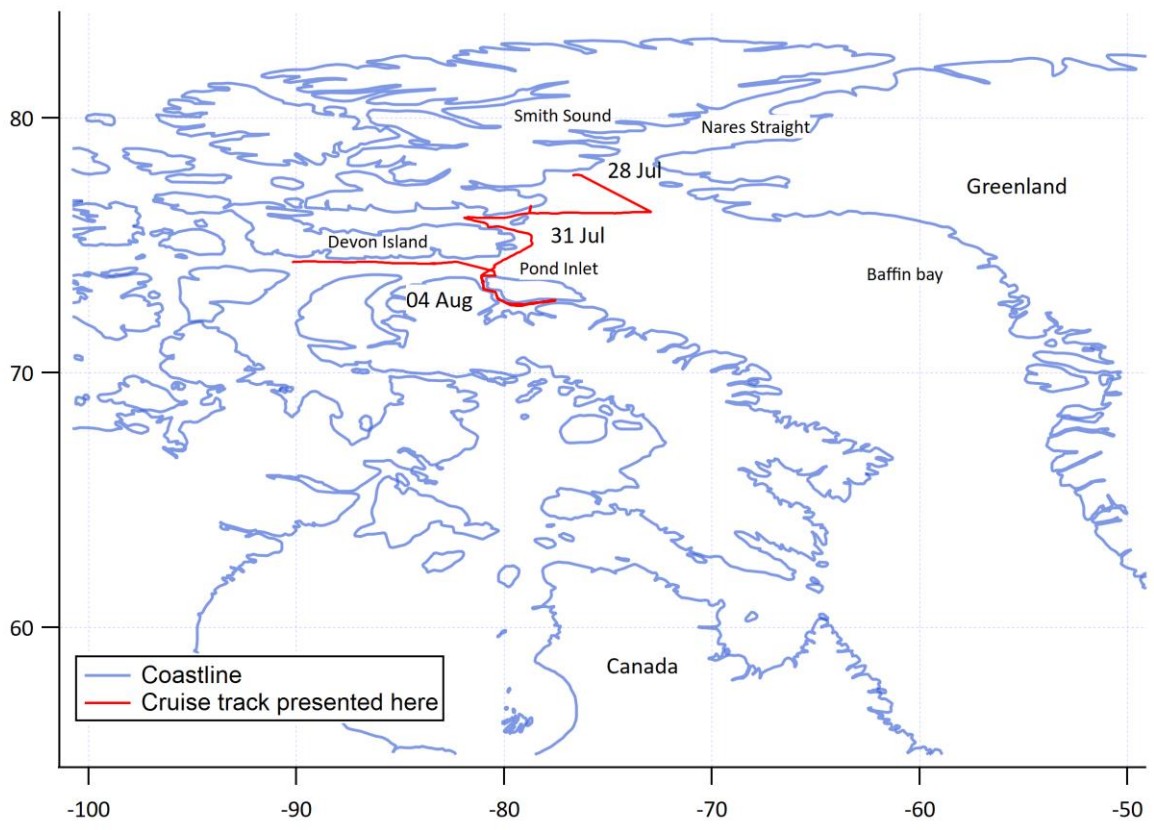

**Figure C1: Cruise track of the data presented here.**





**Table E1: Table listing experimentally determined Henry solubilities of methanol, acetone and acetaldehyde listed in R. Sander (2015)**
**along with the in-text reference and the computed slope of the response in the equilibrator in ppb nmol$^{-1}$ dm$^3$. For full reference of the cited solubilities, please refer to R. Sander (2015). Experimentally determined calibration slope for methanol, acetone and acetaldehyde were 0.00786 ±0.00115 ppb nmol$^{-1}$ dm$^3$, 0.0469 ±0.0145 ppb nmol$^{-1}$ dm$^3$ and 0.0743 ±0.0190 ppb nmol$^{-1}$ dm$^3$.**

| Reference | Henry solubility | Predicted slope ppb nmol$^{-1}$ dm$^3$ |
|---|---|---|
| **Methanol** | | |
| 1. Li et al., (1993) | 7378 | 0.00326 |
| 2. Snider and Dawson (1985) | 7220 | 0.00333 |
| 3. Rytting et al., (1978) | 7378 | 0.00326 |
| 4. Brunett et al., (1963) | 7714 | 0.00312 |
| 5. Glew and Moelwyn-Hughes (1953) | 7430 | 0.00324 |
| 6. Butler et al., (1935) | 7714 | 0.00312 |
| 7. Vitenberg and Dobryakov (2008) | 7044 | 0.00341 |
| 8. St.Pierre et al., (2014) | 2212 | 0.01090 |
| 9. Helburn et al., (2008) | 2616 | 0.00919 |
| 10. Teja et al., (2001) | 6716 | 0.00358 |
| 11. Zhou et al., (2000) | 8882 | 0.00271 |
| 12. Gupta et al., (2000) | 6678 | 0.00360 |
| 13. Altschuh et al., (1999) | 5367 | 0.00448 |
| 14. S. P. Sander et al., (2011) | 6715 | 0.00358 |
| **Acetone** | | |
| 15. Benkelberg et al., (1995) | 891 | 0.0269 |
| 16. Hoff et al., (1993) | 878 | 0.0274 |
| 17. Zhou and Mopper (1990) | 1060 | 0.0227 |
| 18. Guitart et al., (1989) | 746 | 0.0322 |
| 19. Hellmann et al., (1987) | 341 | 0.0703 |
| 20. Snider and Dawson (1985) | 802 | 0.0299 |
| 21. Schoene and Steinhanses (1985) | 1062 | 0.0226 |
| 22. Sato and Nakajima (1979) | 933 | 0.0258 |
| 23. Vittenberg et al., (1975) | 813 | 0.0295 |
| 24. Poulain et al., (2010) | 946 | 0.0254 |
| 25. Ji and Evans (2007) | 863 | 0.0278 |
| 26. Falabella et al., (2006) | 744 | 0.0323 |
| 27. Strekowski and George (2005) | 914 | 0.0263 |
| 28. Straver and de Loos (2005) | 781 | 0.0308 |
| 29. Chai et al., (2005) | 748 | 0.0321 |
| 30. Ayuttaya et al., (2001) (EPICS method) | 325 | 0.0737 |
| 31. Ayuttaya et al., (2001) (static cell, linear form) | 3.0587 | 5.93 |
| 32. Ayuttaya et al., (2001) (direct phase concentration method) | 1725 | 0.0139 |
| 33. S. P. Sander et al., (2015) | 901 | 0.0267 |
| **Acetaldehyde** | | |
| 34. Ji and Evans (2007) | 527 | 0.0455 |
| 35. Straver and de Loos (2005) | 374 | 0.0641 |
| 36. Marin et al., (1999) | 510 | 0.0470 |
| 37. Benkelberg et al., (1995) | 439 | 0.0547 |
| 38. Zhou and Mopper (1990) | 552 | 0.0435 |
| 39. Guitart et al., (1989) | 242 | 0.0991 |
| 40. Betterton and Hoffmann (1988) | 419 | 0.0572 |
| 41. Snider and Dawson (1985) | 408 | 0.0589 |
| 42. Vitenberg et al., (1974) | 298 | 0.0991 |



| 40. Betterton and Hoffmann (1988) | 419 | 0.0572 |
| 41. Snider and Dawson (1985) | 408 | 0.0589 |
| 42. Vitenberg et al., (1974) | 298 | 0.0804 |
| 43. Buttery et al., (1969) | 487 | 0.0493 |
| 44. S. P. Sander et al., (2011) | 444 | 0.0541 |

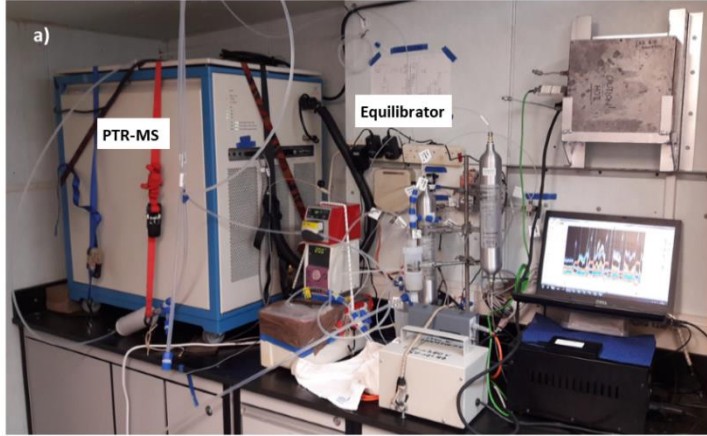
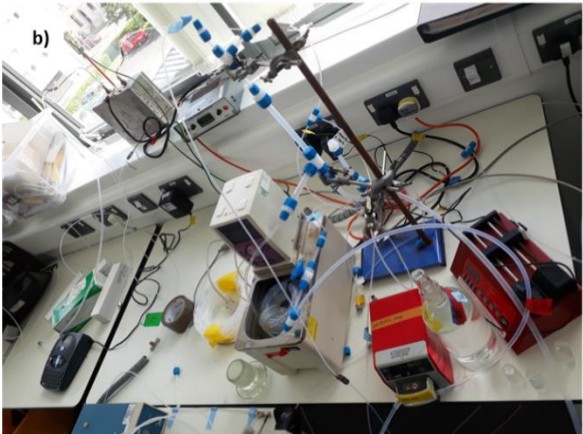

**Figure F1: a) Photograph of the instrument setup during deployment on the CCGS *Amundsen* with the jar trap. b) Photograph of the equilibrator in the laboratory post-deployment with the PTFE tee fitting mounted to separate air and water at the end of the segmented flow tube.**