# Peer review of "Segmented flow coil equilibrator coupled to a Proton Transfer Reaction Mass Spectrometer for measurements of a broad range of Volatile Organic Compounds in seawater"

_Ocean Science, 2019_

## Referee Comment (RC1) · Anonymous Referee #1 · 21 Mar 2019

Review of Segmented flow coil equilibrator coupled to a Proton Transfer Reaction Mass Spectrometer for measurements of a broad range of Volatile Organic Compounds in seawater By Wohl, et. al. Ocean Sci. Discuss., https://doi.org/10.5194/os-2019-5

Please see attached pdf of formatted review.

In this manuscript, Wohl et. al demonstrate the application of a Segmented Flow Coil Equilibrator (SFCE) for VOC and OVOC online dissolved gas analysis using Proton Transfer Reaction Mass Spectrometry (PTR-MS). The performance of the combined

system is demonstrated through a comprehensive suite of calibration, background, and response time tests. This manuscript's main contribution to the field is the expansion of our knowledge behaviors of the SFCE dissolved gas sampling method and aspects and pitfalls of successful implementation. Of particular interest is the simplicity of the design combined with the extent and speed of equilibration of both low and high solubility gases. This information is of interest to both the ocean science community, but also air quality, ecology, and inland water quality applications, and represents simple alternative to other equilibrator designs. Additionally, there is a lot of interesting data regarding the operation of the PTR-MS, including ion source and drift tube tuning that are valuable insights to the PTR-MS user and science community. Overall, this documentation of a sound instrument development project and the results are very exciting and well demonstrated. Additionally, the authors have thoughtfully included detailed appendixes/SI that clearly lays out many details of their work.

There are several broad areas where the manuscript could use improvement. (1) Nomenclature. Many colloquialisms are used that imprecisely describe materials and processes at in question, which obfuscates the discussion but also harms communication across fields. a. Using the IUPAC Glossary of Terms Related to Solubility (10.1351/pac200880020233) as a reference i. Instead of 'airside' use 'gas phase' ii. Instead of 'waterside' use 'dissolved gas concentration' b. While softer/harder ionization is technically correct, it's a lot more illuminating to discuss proton affinity differences and effective temperatures, which are the forces at play in the PTR-MS drift tube and ion optic system. c. The protonated target molecule is the "primary ion". A charged fragment of dissociation should be called a "product ion" or "fragment ion".

(2) Uncertainty. Consistently state and propagate uncertainty and significant figures. a. Section 3.2 needs attention. The reader cannot determine the input concentrations for the evasion experiments with the information provided. Are the purge factors really known to the stated (per-mil) precision?

(3) Harmonize section 3 and 4. These sections seem a bit repetitive and scattered, conceptually jumping back and forth between PTR-MS and SFCE tests. I suggest moving the theoretical/math into Section 3, and calling it "Derivation of Dissolved VOC concentrations from SFCE/PTR-MS measurements". The experimental/operational work (3.1 Determination of System Background and 3.2 Estimation of Equilibration Efficiency) could be moved into section 4. Section 3.2 and 4.2 seem like they could be combined. Another thought is that the SFCE testing is largely disconnected from the PTR-MS humidity and calibration testing, so those phases could each get their own sections (Section 3: "PTR-MS operation", Section 4 "SFCE testing and operation")

(4) Since the manuscript deals with both gas phase mixing ratios and dissolved concentrations, I suggest using "ppbv" instead of "ppb", as gas mixing ratios are typically by molar volume while aqueous mixing ratios are often by mass.

(5) Instead of a long series of appendixes, could that information just be put in the supplemental material? Using the PTR-MS at 160 Td yields some unusual data, but in this case PTR-MS is fundamentally just the detector and the main focus of the manuscript is the SFCE application.

Specific comments and suggestions:

Line 16: 1 min instead of 1min.

Line 43-60: Are the authors aware of any investigators using hollow fiber membrane contactors for online dissolved gas analysis in seawater? (I am not aware of any example, but they are popular in inland surface water, groundwater, and industrial settings. So perhaps there is an example that escapes my limited search and knowledge?)

Line 90: Consider rephrasing to "In this paper we extend the application of the segmented flow coil equilibrator..." The core design is substantially similar, but the target analytes are hitherto undocumented. 120-135 and 195-220: I note that the inlet water is warmed to 20 °C. I wondered how much N2 would exsolve from the water and add to the total gas flow, as this would effectively dilute the measured VOCs. Air is about 25%

less soluble at 20 °C than at 0 °C, and using the solubilities of O2 and N2 as proxies, it seems like the amount of air exsolved from a 100 cm3/min flow of water warming from 0 to 20 °C would be about 0.5 sccm: so the temperature change is not causing enough off gassing to measurably modify the mixing ratios measured in the equilibrator.

Line 197: It would be more clear to write "R=8.314 ïĆť 10-3 dm3 Pa mole-1 K-1" or similar.

Line 260-270: PTFE has a measurable permeability to many gases, and at thin cross-sections, is used as a membrane material, leveraging that property. Looking at some manufacturer datasheets, acetone and methanol are among the most permeable gases in PTFE. It seems like switching to the PTFE tee fitting improves the situation by reducing residence times of gas/fluid and minimizing unswept volumes. Would you recommend stainless steel or glass for future designs?

Line 260: Can you give us an idea of what volume of water was in the PTFE jar and tee at steady state here? That would help give us an idea of water residence time in the entire system. (Figure 1 gives a hint about the tee seems like about 10 cm3, but the jar is unknown.)

Line 285-295: Can you include some more detail, or perhaps expand the appendix/SI to include more specific information about how the evasion standards were made and used. How much MilliQ water was used? What was the pipette volume/precision? What was the dilution volume/mass and precision? How many dilutions were done to get to the final stock? How long was the SFCE purged before measurement? Line 305-310: Are the solubilities known to a level of accuracy that allow for 5 significant figures? If not, perhaps the uncertainty should be clarified.

Line 354: "100cm3 n: 100cm3" should this be restated as "air and water at equal flow rates of 100 cm3 at 20 C"?

Line 370: Peristaltic pumps are notoriously bad actors in dissolved gas sampling, and

require assiduous attention to maintain constant flow over time. Would you recommend another pump, perhaps a magnetically coupled stainless steel gear pump, to others?

Line 376: "Our aim is to build an equilibrator that fully equilibrates for the very soluble OVOCs". This sentence succinctly describes the manuscript. Consider if it can be placed somewhere in the abstract or introduction (perhaps around line 93).

Line 516: Hollow cathode DC plasma discharge

Line 528: Instead of (H218O+)H2O (which would be m/z 39) you probably mean (H216O)H3O+.

Line 530: A great deal is written here about how much effort is put into managing humidity to achieve consistent results. Getting a handle on these relationships is a curcial aspect for achieve maximal PTR-MS performance and is both widely recognized and documented from a very early point in the PTR-MS methods arrival. The implementation as described basically has a PTR-MS with a heated inlet and vacuum system, in a conditioned space aboard a ship, drawing a gas/water mixture through a temperature-controlled coil (the SFCE) at 20C. The vapor pressure of water at this temperature is around 17 torr. The flow rate of water vapor through the PTR-MS ion source was essentially constant (3 sccm). One might surmise these measurements benefited from an extremely predictable and stable input of water relative to air quality and biogeochemical measurements. How much variation in drift tube humidity was there? Can you show us a plot of % m/z 37 over time? How about %m/z 55? What's the return of this tweaking vs running the PTR-MS in a more conventional manner?

Line 544: "ionization by water clusters is lower energy and..."

Line 544-560: Running the instrument at 160 Td is unusual, as is the water flow (5 sccm) and discharge current (3mA). Most investigators report a sweet spot between 100 and 140 Td, with resulting uncertainties in the range of 5-25% RSD. While discharge conditions are not as commonly reported (to my dismay), the latest HS-PTR-

MS user manuals up to 2011 (the last I have access to) suggest water flow rates between 6-15 sccm and discharge of 4-6 mA. While it does have the effect of reducing the abundance of hydrated clusters in the drift tube, it also decreases the reaction time and greatly increases fragmentation, both of the target analytes and of higher mass molecules, from whom the fragment ions can then interfere with the measurements. There are basically two selection criteria of the PTR-MS method (1) Only molecules with a proton affinity higher than water are detected (2) Those protonated ions can be uniquely detected at a specific m/z ratio either directly or by some signal deconvolution. By operating the PTR-MS in this configuration, it's likely that those conditions are only true for a select set of compounds. I would surmise that performance with monoterpenes, acetic acid, and anything with a terminal hydroxyl group to be especially problematic. The high degree of fragmentation of isoprene observed here is emblematic of these operating conditions. The authors should emphasize that in seeking to suppress cluster formation in the drift tube, they are making substantial performance tradeoffs in other areas.

Line 570-603: I'm a bit confused: how much of the background signal of OVOCs are being attributed to humidity and how much do you think is from OVOCs in the water can? Can you comment on the background signal of these other OVOCs over time? Water held under dynamic vacuum preferentially degases, so one would expect any dissolved gases to be removed from the water can after a prolonged period of PTR-MS operation, especially in a warm instrument cabinet on a rocking ship, turning over the water. Reviewing several years of my own PTR-MS datasets, I see elevated backgrounds immediately after the instrument is turned on after service/water can fills, but they quickly recede to a stable signal (usually a few hundred CPS) with an extremely weak relationship between m/z 37 and m/z 45 or m/z 63.

Line 651: I suggest calling Appendix E: "Compilation of published solubilities for methanol, acetone, and acetaldehyde".

Table 3: (and throughout). For consistency, I suggest sticking with nmol/dm-3 throughout, and using scientific notation for isoprene instead of pmol dm-3. i.e. 9.96 * 10-3 +/-
1.25 * 10-3 nmol dm-3)

Figure 3 (a): Typo "assuming" not "assuning".

Figure 4: I suggest either using all black or using some color variation. It's hard to tell
the 1:1 line and the fit to the measurement lines.

Figure 5: This plot and caption could use some clarification. This is a comparison
of range solubilities observed with the SFCE-PTR-MS system and values predicted
from literature. The meaning of the numbers in the legends (1-44) of Figure 5 are not
immediately clear. To help, each line could be "Ref. x" (x=1-44), with "This work" as
the thick red line and "S. P. Sander" as the thick blue one. In the caption, please write
what you want the reader to take away from this demonstration. It seems like you are
seeing lower solubility than the literature values.

Figure 7: Can this be remade as a full page plot? The horizontal axis is extremely tight.
If size is an issue, plot gases of similar magnitude on the same subplot and use the right
axis. I suggest adding to the x axis "Sample Date & Time ( HH:MM DD/MM/YYYY)".

Figure B-1: Are there error bars (like the caption says) in these plots? They are not
rendering on my printer or pdf.

Please also note the supplement to this comment:
https://www.ocean-sci-discuss.net/os-2019-5/os-2019-5-RC1-supplement.pdf

---

## Referee Comment (RC2) · Anonymous Referee #2 · 22 Mar 2019

General comments: In this paper, the authors developed and evaluated a novel measurement system, SFCE extraction coupled with PTR-MS. The authors also deployed the system on a research cruise in the Canadian Arctic and obtained suitable dataset of dissolved VOC in surface water. The success of the reduction of sample water volume made the availability of this system for not only continuous measurement but also discrete sample, therefore, vertical profiling of dissolved VOCs. This function has not reported suitably in the previous continuous measurement systems. Generally, the explanation of the system and their evaluation of the system has been well organized and

presented in this paper, and I totally agree with their arguments here. I found that the improvement shown in this manuscript is worthy to be published in Ocean Science with minor revision.

Specific comments: References: Please check the form of references shown in the text. For example, "Blando & Turpin, 2000" (page 1, line 28) should be shown as "Blando and Turpin, 2000", and "de Bruyn, Clark, Senstad, Barashy, & Hok, 2017" (page 2, line 43) should be shown as "de Bruyn et al., 2017".

Page 2, line 35 Please add the suitable references.

Page 3, line 88 The response time of 10 min is only for isoprene, not for the other compounds such as acetone, methanol and so on.

Page 3, line93-102 Those explanations itself are generally well, but not suitable in introduction section. I assume that those sentences could be removed.

Page 10, line 334 Ionized toluene should be found in not only m/z 93 but also m/z 79 depending on the drift tube voltage. Did the authors find the fragmentation of toluene? m/z 79 is used to identify benzene amount, therefore, the authors need to care of existence of the fragmentation.

Page 11, line 354 What is the ratio of "100 cm3n:100 cm3" here? Maybe typo?

---

## Author Comment (AC1) · 10 May 2019

**Reply to Reviewer 1 comments for: Segmented flow coil equilibrator coupled to a Proton Transfer Reaction Mass Spectrometer for measurements of a broad range of Volatile Organic Compounds in seawater**

Many thanks for the thoughtful comments from this anonymous reviewer. Thank you for your time spotting some of the nomenclature errors and typos. The reviewer has been able to provide thought provoking comments and point out some of the trade-offs in our setup. Please see our responses below. Reviewer comments are in italic and author's replies can be found in normal font. The changes to the manuscript are presented as figures taken from the manuscript with the changes made indicated by red track changes.

**General comments**

*(1)Nomenclature. Many colloquialisms are used that imprecisely describe materials and processes at in question, which obfuscates the discussion but also harms communication across fields. a. Using the IUPAC Glossary of Terms Related to Solubility (10.1351/pac200880020233) as a reference i. Instead of 'airside' use 'gas phase' ii. Instead of 'waterside' use 'dissolved gas concentration' b. While softer/harder ionization is technically correct, it's a lot more illuminating to discuss proton affinity differences and effective temperatures, which are the forces at play in the PTR-MS drift tube and ion optic system. c. The protonated target molecule is the "primary ion". A charged fragment of dissociation should be called a "product ion" or "fragment ion".*

Thank you for this comment.

    a.   Solubility terms
    i.       Suggestion accepted
    ii.     Suggestion accepted, except lines 300 and 630 where "dissolved gas concentration" did not fit well and was replaced by "seawater" instead. See below.

> For evasion experiments, a solubility-dependent fraction of dissolved VOCs is transferred into the gas phase during the equilibration process. Thus, the final dissolved concentration will be somewhat lower than the
> 300   initial concentrations. To account for the removal of a fraction of these gases from the  seawater during equilibration, a purging factor (PF) based on mass conservation is applied. The PF is the ratio between the dissolved gas  concentration before and after complete equilibration in the coil. The derivation of this compound specific purging factor is presented in the appendix. At equal air and water flow rates, it simplifies to:

$$PF = \frac{C_w(before\ equilibration)}{C_w(after\ equilibration)} = \frac{1}{H} + 1 \qquad (4)$$

    b.   Suggestion accepted, see below the applied changes to the manuscript

> Humidity has several potential effects on the measurement: (i) The additional water molecule stabilises the primary ion by sharing the positive charge thus increasing its proton affinity (Blake et al., 2009). For example benzene and toluene possess intermediate proton affinities and are ionised by the primary ion, but not the water cluster (Warneke et al., 2001). On the other hand, this process decreases the proton affinity difference between the primary ion and the VOCs and  so the excess energy released on proton transfer reaction (de Gouw and& Warneke, 2007). This leads to less fragmentation for example of isoprene in the drift tube (Schwarz et al. 2009).; (ii) humidity in
>
> …

For isoprene, the opposite effect is observed (Fig. B4) where the stabilising water cluster from humidity

615 decreases the excess energy released upon ionisation thus increasing the yield of the parent ion at m/z 69. Note that humidity dependant fragmentation of isoprene in PTR-MS has been observed before (Schwarz et al. 2009). Other masses that isoprene fragments can be found are m/z 39 and m/z 41.

c.  Suggestion accepted, see below the applied changes to the manuscript

Humidity has several effects on the measurement: (i) The additional water molecule stabilises the primary ion by sharing the positive charge thus increasing its proton affinity (Blake et al. 2009). For example benzene and toluene possess intermediate proton affinities and are ionised by the primary ion, but not the water cluster

535 (Warneke et al. 2001). On the other hand, this process decreases the proton affinity difference between the primary ion and the VOC and thus the excess energy released on proton transfer reaction . This leads to less fragmentation for example of isoprene in the drift tube (Schwarz et al. 2009); (ii) humidity in the drift tube leads to non-collision rate limited reactions; (iii) large water

…

615         For isoprene, the opposite effect is observed (Fig. B4) where the stabilising water cluster from humidity decreases the excess energy released upon ionisation thus increasing the yield of the primary ion at m/z 69. Note that humidity dependant fragmentation of isoprene in PTR-MS has been observed before (Schwarz et al. 2009). Other masses that isoprene fragment ions can be found are m/z 39 and m/z 41.

*(2) Uncertainty. Consistently state and propagate uncertainty and significant figures.*
*a. Section 3.2 needs attention. The reader cannot determine the input concentrations*
*for the evasion experiments with the information provided. Are the purge factors really*
*known to the stated (per-mil) precision?*

Suggestion accepted, see below the applied changes to the manuscript

Two methods are used to assess the equilibration efficiency of the SFCE: evasion and invasion.  In evasion experiments, liquid standards of methanol, acetone and acetaldehyde were prepared by serial dilution of the pure solvent in the same batch of MilliQ water. Aliquots of pure, undiluted methanol (For spectroscopy Uvasol) and acetone (HPLC standard) were dispensed using volumetric pipettes. A 1 cm³ volumetric flask was used to aliquot pure acetaldehyde (>=99.5%, A.C.S. Reagent). Subsequent dilutions utilised a volumetric pipette and volumetric flask to prepare liquid standards ranging from 3 to 30 nM for acetone and acetaldehyde and 30 to 300 nM for methanol. Liquid standards of Isoprene and DMS were prepared gravimetrically airtight each day. A syringe pump (New Era Pump Systems) was used to dynamically dilute DMS and isoprene standards in a flow of MilliQ water. This yielded DMS standards of up to 7 nM and isoprene standards of up to 2 nM. For this calibration, the flow rate of MilliQ water is measured at the drain. In evasion calibrations, a solubility-dependent fraction of dissolved VOCs is transferred into the gas phase during the equilibration process. Thus, the final dissolved concentration will be somewhat lower than the initial concentrations. To

The precision of the purging factor depends on the precision of the solubility measurement. Since we report solubilities with two significant figures, we decide to report the purging factor with two significant figures as well.

For freshwater, computed purging factors assuming full equilibration and equal zero air: water flows are: 1.000 for methanol, 1.001 for acetone, 1.002 for acetaldehyde, 1.06 for DMS, 1.18 for benzene, 1.21 for toluene and 2.57 for isoprene. The same computation in seawater gives the following purging factors: 1.000 for methanol, 1.001 for acetone, 1.002 for acetaldehyde, 1.08 for DMS, 1.22 for

310 benzene, 1.26 for toluene and 2.96 for isoprene. We see that PF varies from being insignificant (~=1) for highly soluble VOCs to quite large (>>1) for the sparingly soluble gases. To compute the expected headspace

*(3) Harmonize section 3 and 4. These sections seem a bit repetitive and scattered, conceptually jumping back and forth between PTR-MS and SFCE tests. I suggest moving the theoretical/math into Section 3, and calling it "Derivation of Dissolved VOC concentrations from SFCE/PTR-MS*

*measurements". The experimental/operational work (3.1 Determination of System Background and 3.2 Estimation of Equilibration Efficiency) could be moved into section 4. Section 3.2 and 4.2 seem like they could be combined. Another thought is that the SFCE testing is largely disconnected from the PTR-MS humidity and calibration testing, so those phases could each get their own sections (Section 3: "PTR-MS operation", Section 4 "SFCE testing and operation").*

> Suggestion accepted. Section 2.2 Field deployment and section 5 Field testing have been merged to one section 5 at the end of the manuscript. Following the comments, section 4.1. Effect of humidity on the PTR-MS measurements has been moved to a new section 2.2 entitled PTR-MS operation, thus removing the operation/testing of the PTR-MS detector from the description of the SFCE equilibration system. Section 3 and 4 have been renamed according to the reviewer's comments. Further following the reviewer's comments, section 3.2 estimation of equilibration efficiency has been moved to a new section 4.1. This has been combined with section 4.2 Measurement sensitivity toward air:water flow ratio and presumably also 4.3 Equilibration efficiency. In this merged section, new subsections have been made to avoid an overwhelmingly large and unstructured section.

*(4) Since the manuscript deals with both gas phase mixing ratios and dissolved concentrations, I suggest using "ppbv" instead of "ppb", as gas mixing ratios are typically by molar volume while aqueous mixing ratios are often by mass.*

> Suggestion accepted, including in the figure notations

*(5) Instead of a long series of appendixes, could that information just be put in the supplemental material? Using the PTR-MS at 160 Td yields some unusual data, but in this case PTR-MS is fundamentally just the detector and the main focus of the manuscript is the SFCE application.*

> Suggestion accepted. The appendix will be included as supplemental information instead. Thank you for your feedback. We have deployed PTR-MS at 150 Td during a more recent campaign. The high Td value is mainly due to the higher than ambient drift tube temperature (80 deg C) in our setup, which was recommended by the manufacture for measurements of these VOCs.

**Specific comments and suggestions:**

*Line 16: 1 min instead of 1min.*

Suggestion accepted

*Line 43-60: Are the authors aware of any investigators using hollow fiber membrane contactors for online dissolved gas analysis in seawater? (I am not aware of any example, but they are popular in inland surface water, groundwater, and industrial settings. So perhaps there is an example that escapes my limited search and knowledge?)*

Thank you for pointing this out. We added a few examples in the section where membrane equilibrators are discussed in the relevant section of the introduction.

75 from the showerhead lengthens the equilibrator's response time for highly soluble gases, making it less suitable for high frequency measurements of highly soluble VOCs such as methanol (Kameyama et al., 2010). Membrane equilibrators avoid spray formation and allow for selective diffusion. Hollow fibre membranes have previously been used for measurement of dissolved $CO_2$ (Hales et al., , 2005; Sims et al., 2017) and DMS (Tortell, 2005; Yang et al., 2011)

80  By using a hydrophobic

*Line 90: Consider rephrasing to "In this paper we extend the application of the segmented flow coil equilibrator …" The core design is substantially similar, but the target analytes are hitherto undocumented. 120-135 and 195-220: I note that the inlet water is warmed to 20 °C. I wondered how much N2 would exsolve from the water and add to the total gas flow, as this would effectively dilute the measured VOCs. Air is about 25% less soluble at 20 °C than at 0 °C, and using the solubilities of O2 and N2 as proxies, it seems like the amount of air exsolved from a 100 cm3/min flow of water warming from 0 to 20 °C would be about 0.5 sccm: so the temperature change is not causing enough off gassing to measurably modify the mixing ratios measured in the equilibrator.*

Suggestion accepted. As the reviewer stated, dilution due to N2 exsolving with warming does not significantly affect the VOC concentrations measured.

In this paper we extend the application of the segmented flow coil equilibrator (SFCE). It is adopted from the designs used by Xie et al. (2001) and Blomquist et al. (2017) for measurements of carbon monoxide and DMS, respectively.  We couple this

95 equilibrator to a PTR-MS with the settings optimised for measurement of VOCs in the water phase. The main

*Line 197: It would be more clear to write "R=8.314 ïC´ t' 10-3 dm3 Pa mole-1 K-1" or similar.*

Suggestion accepted

Where $n$ (mol) represents the quantity of matter, $V$ ($dm^3$) represents the volume of gas, $P$ (Pa) represents the

200 pressure, $R$ = 8.314 ($m^3$ Pa $K^{-1}$ $mol^{-1}$) and $T$= 293.15 (K). A conversion factor of 0.001 is applied to convert from $m^3$ to $dm^3$.

*Line 260-270: PTFE has a measurable permeability to many gases, and at thin cross sections, is used as a membrane material, leveraging that property. Looking at some manufacturer datasheets, acetone and methanol are among the most permeable gases in PTFE. It seems like switching to the PTFE tee fitting improves the situation by reducing residence times of gas/fluid and minimizing unswept volumes. Would you recommend stainless steel or glass for future designs?*

Thank you for this advice. We note our system is slightly overpressured, such that contamination from lab air should not occur even if PTFE is slightly permeable towards OVOCs. The effect of different materials for OVOC measurements has been investigated for example on methanol (Beale et al,. 2011, supplementary material figure S1). They found that methanol strongly absorbs on the walls of stainless-steel tubing.  Albeit possibly costly, glass may be a good idea indeed. However, this would make ship board deployment more complicated due to the fragility of glass.

*Line 260: Can you give us an idea of what volume of water was in the PTFE jar and tee at steady state here? That would help give us an idea of water residence time in the entire system. (Figure 1 gives a hint about the tee seems like about 10 cm3, but*

*the jar is unknown.)*

Suggestion accepted.

The final blank we determined was the "wet equilibrator" blank.  This consisted of stopping the water flow into the equilibrator and purging the wet equilibrator (that had been coated with bottom seawater) with zero air for 20 min. During this blank measurement, humidity in the headspace remained constant as small water droplets remained inside of the coil and were not substantially dried by the zero air.  During the Arctic cruise, the wet equilibrator blank consistently resulted in the lowest blank reading on the PTR-MS for all VOCs except for methanol and acetone as a result of a contamination (discussed below). Thus, in practice the wet equilibrator blank seems to be the best surrogate for a "true" water blank for almost all VOCs measured here. During the wet equilibrator blank, the bottom of the PTFE jar or PTFE tee is filled with approximately 5 mL of seawater. In the case of the tee, the water leaves the tee approximately immediately. During measurement of seawater, the residence time of zero air and seawater in the equilibrator is approximately 1 min. The residence time of zero air in the wet equilibrator blank doubles during the blank.

*Line 285-295: Can you include some more detail, or perhaps expand the appendix/SI to include more specific information about how the evasion standards were made and used. How much MilliQ water was used? What was the pipette volume/precision? What was the dilution volume/mass and precision? How many dilutions were done to get to the final stock? How long was the SFCE purged before measurement?*

Thank you for this suggestion. A paragraph addressing these questions has been added to supplementary material E.

Here we provide more detail on how the evasion standards of methanol, acetone and acetaldehyde were prepared in MilliQ water. For this, 303 mm$^3$ of pure methanol (For spectroscopy Uvasol) and 55 mm$^3$ acetone (HPLC standard) were diluted in a 0.5 dm$^3$ volumetric flask labelled as "A". In a 1 dm$^3$ volumetric flask labelled "B", 1 cm$^3$ acetaldehyde was dissolved as measured out using a 1 cm$^3$ volumetric flask (>=99.5%, A.C.S. Reagent). A third flask labelled "C" of 0.5 dm$^3$ was used to further dilute 330 mm$^3$ of flask "A" and 330 mm$^3$ of flask "B". Different amounts of the flask labelled "C" were dissolved in 800 cm$^3$ sampling bottles filled with MilliQ water. Standards were typically analysed within 4 h of dissolving the pure OVOC in water. The same 10 dm$^3$ batch of MilliQ water was used to dissolve the pure standards and it was also syphoned into the sampling bottles. A MilliQ blank has been subtracted from the measurements of the evasion calibration curves. The same three air displacement micropipettes (20-200 mm$^3$, 100-1000 mm$^3$, 0.5-5 cm$^3$) with plastic tips were used for this dilution. All volumetric flasks were class A volumetric glassware. The SFCE was typically purged with MilliQ water for at least 30 min before starting measurement. The solubilities of the VOCs at 20 ° C used to compute the expected mixing ratio are presented in table E1.

*Line 305-310: Are the solubilities known to a level of accuracy that allow for 5 significant figures? If not, perhaps the uncertainty should be clarified.*

Suggestion accepted

The precision of the purging factor depends on the precision of the solubility measurement. Since solubilities are reported in this paper to two significant figures, purging factor is reported here to two significant figures as well. For freshwater, computed purging factors assuming full equilibration and equal zero air: water flows are: 1.00 for methanol, 1.00 for acetone, 1.00 for acetaldehyde, 1.06 for DMS, 1.18 for

320    benzene, 1.2 for toluene and 2.5 for isoprene. The same computation in seawater gives the following purging factors: 1.00 for methanol, 1.00 for acetone, 1.00 for acetaldehyde, 1.08 for DMS, 1.22 for benzene, 1.2 for toluene and 2.96 for isoprene. We see that PF varies from being insignificant (~=1) for highly soluble VOCs to quite large (>>1) for the sparingly soluble gases. To compute the expected headspace

*Line 354: "100cm3 n: 100cm3" should this be restated as "air and water at equal flow rates of 100 cm3 at 20 C"?*

Suggestion accepted.

*Air and water at equal flow rates of 100 cm³ at 20 °C are*  chosen to allow for a reasonably long equilibration time, large surface area for exchange, and so high signal while
370    satisfying the air flow requirements of the PTR-MS. They are also chosen such that the stripping of the soluble

*Line 370: Peristaltic pumps are notoriously bad actors in dissolved gas sampling, and require assiduous attention to maintain constant flow over time. Would you recommend another pump, perhaps a magnetically coupled stainless steel gear pump, to others?*

Thank you very much for your recommendation. We are looking forward to taking your recommendation on board.

*Line 376: "Our aim is to build an equilibrator that fully equilibrates for the very soluble OVOCs". This sentence succinctly describes the manuscript. Consider if it can be placed somewhere in the abstract or introduction (perhaps around line 93).*

Suggestion accepted. This sentence has been moved from line 376 to line 96.

In this paper we extend the application of the segmented flow coil equilibrator (SFCE). It is adopted from the designs used by Xie et al. (2001) and Blomquist et al. (2017) for measurements of carbon monoxide and DMS, respectively. We couple this
95    equilibrator to a PTR-MS with the settings optimised for measurement of VOCs in the water phase. Our aim is to build an equilibrator that fully equilibrates for the very soluble OVOCs.

*Line 516: Hollow cathode DC plasma discharge*

Suggestion accepted.

To measure the VOC concentrations, we use a commercially available high sensitivity Proton-Transfer-Reaction

530    Mass Spectrometer (de Gouw & Warneke, 2007; Lindinger & Jordan, 1998). Briefly, water vapor is ionised in a *Hollow cathode DC plasma discharge*. The hydronium ions react with sample air in the drift tube.

*Line 528: Instead of (H218O+)H2O (which would be m/z 39) you probably mean (H216O)H3O+.*

    Indeed. Suggestion accepted.

water clusters. In practice, such water dimmers are monitored at m/z 37 (i.e. isotopic hydronium water cluster $(H_2^{18}O^+)H_2O$) as a percentage of the primary ion count, accounting for isotopic abundance (Blake et al. 2009):

$$H_2O + H_3O^+ \rightarrow H_3O^+(H_2O). \tag{A1}$$

*Line 530: A great deal is written here about how much effort is put into managing humidity to achieve consistent results. Getting a handle on these relationships is a curcial aspect for achieve maximal PTR-MS performance and is both widely recognized and documented from a very early point in the PTR-MS methods arrival. The implementation as described basically has a PTR-MS with a heated inlet and vacuum system, in a conditioned space aboard a ship, drawing a gas/water mixture through a temperature controlled coil (the SFCE) at 20C. The vapor pressure of water at this temperature is around 17 torr. The flow rate of water vapor through the PTR-MS ion source was essentially constant (3 sccm). One might surmise these measurements benefited from an extremely predictable and stable input of water relative to air quality and biogeochemical measurements. How much variation in drift tube humidity was there? Can you show us a plot of % m/z 37 over time? How about %m/z 55? What's the return of this tweaking vs running the PTR-MS in a more conventional manner?*

    Thank you for this comment. Figure 1 shows a timeseries of the drift tube humidity as monitored as a fraction of m/z 21 during the deployment in the Canadian Arctic. The figure shows the drift tube humidity while measuring equilibrator headspace, outside air and zero air from a gas cylinder.

[Figure]

*Figure 1: Timeseries of the drift tube humidity during the deployment in the Canadian Arctic when measuring outside air, equilibrator headspace and zero air from a gas canister.*

Figure 1 shows that the equilibrator headspace humidity (as indicated by m/z 37) was almost always less than 5% of the m/z 21 signal.

Unfortunately, we did not monitor m/z 55, but we expect it to be very small given the small amount of m/z 37 monitored already.

Figure 2 shows the effect of varying the drift tube voltage on the abundance of the water hydronium cluster. The figure shows that at decreasing drift tube voltage, the abundance of hydronium water clusters increases.

[Figure]

*Figure 2: Abundance of water hydronium cluster in the drift tube as a function of the drift tube voltage.*

An abundance above 5% of m/z 37 is undesirable. In a more recent deployment, the drift tube voltage was set to 640V which equates to 147 Td.

*Line 544: "ionization by water clusters is lower energy and …"*

  Suggestion accepted.

550 (Warneke et al. 2001). On the other hand, this process decreases the proton affinity difference between the primary ion and the VOC and thus the excess energy released on proton transfer reaction ionisation by the water cluster is softer and hence. This leads to less fragmentation for example of isoprene in the drift tube (Schwarz et al. 2009); (ii) humidity in the drift tube leads to non-collision rate limited reactions; (iii) large water

*Line 544-560: Running the instrument at 160 Td is unusual, as is the water flow (5 sccm) and discharge current (3mA). Most investigators report a sweet spot between 100 and 140 Td, with resulting uncertainties in the range of 5-25% RSD. While discharge conditions are not as commonly reported (to my dismay), the latest HS-PTR-MS user manuals up to 2011 (the last I have access to) suggest water flow rates between 6-15 sccm and discharge of 4-6 mA. While it does have the effect of reducing the abundance of hydrated clusters in the drift tube, it also decreases the reaction time and greatly increases fragmentation, both of the target analytes and of higher mass molecules, from whom the fragment ions can then interfere with the measurements. There are basically two selection criteria of the PTR-MS method (1) Only molecules with a proton affinity higher than water are detected (2) Those protonated ions can be uniquely detected at a specific m/z ratio either directly or by some signal deconvolution. By operating the PTR-MS in this configuration, it's likely that those conditions are only true for a select set of compounds. I would surmise that performance with monoterpenes, acetic acid, and anything with a terminal hydroxyl group to be especially problematic. The high degree of fragmentation of isoprene observed here is emblematic of these operating conditions. The authors should emphasize that in seeking to suppress cluster formation in the drift tube, they are making substantial performance trade-offs in other areas.*

  Many thanks for these thoughts. This is helpful for our future research. The discharge current and the water flow were operated at bespoke settings following recommendations by the manufacturer. We suspect their motivation is to extend the lifetime of the source. We acknowledge

that the high drift tube voltage does lead to some fragmentation of compounds such as isoprene. However, our main focus is measurement of the very small OVOCs that do not tend to fragment. We also acknowledge that the high drift tube voltage does affect the sensitivity of the instrument as it reduces the reaction time in the drift tube. The effect of this should be captured in the gas phase calibrations. As mentioned before, in more recent deployments, the PTR-MS drift tube voltage has been set to 640V. Please see below on how the manuscript was changed upon your recommendation.

Clearly, excessive water clustering in the drift tube is undesirable. To keep the water dimer to be < 5% of the primary ion count when measuring headspace equilibrator, the PTR-MS drift tube was operated at 160Td (700V, 2.2 mBar and 80°C in the drift tube). The water vapor flow into the source was set to 5 $cm^3n$/min, the

575    source current at 3 mA and the source valve to 35%. At these settings, the amount of hydronium water clusters is below 1% when measuring dry zero air and the amount of $O_2^+$ ions is below 0.7% of the primary ion counts. Residual water cluster measured during dry canister measurement is due to unionised water vapor from the hollow cathode entering the drift tube (Warneke et al. 2001). The disadvantage of this high drift tube voltage are increased fragmentation and a reduced reaction time in the drift tube leading to overall lower sensitivity. In

580    this case these were trade-off worth considering since the focus of these measurements are small and generally not fragmenting molecules. The decrease in sensitivity is captured through regular gas phase calibrations. The interested reader looking to do the same measurements should feel free to adapt those settings to their requirements.

*Line 570-603: I'm a bit confused: how much of the background signal of OVOCs are being attributed to humidity and how much do you think is from OVOCs in the water can? Can you comment on the background signal of these other OVOCs over time? Water held under dynamic vacuum preferentially degases, so one would expect any dissolved gases to be removed from the water can after a prolonged period of PTRMS operation, especially in a warm instrument cabinet on a rocking ship, turning over the water. Reviewing several years of my own PTR-MS datasets, I see elevated backgrounds immediately after the instrument is turned on after service/water can fills, but they quickly recede to a stable signal (usually a few hundred CPS) with an extremely weak relationship between m/z 37 and m/z 45 or m/z 63.*

Suggestion accepted. Our results show that for all compounds, except DMS and acetaldehyde, the backgrounds seem independent of sample humidity – i.e. all of the VOC background is coming from the source $H_2O$ reservoir. When measuring dry synthetic air, all of the background can be attributed to VOCs in the water can. Synthetic air measurement for DMS or acetaldehyde is typically below 0.25 ppb. The contribution of sample humidity for both compounds is around 0.6 ppb as seen in fig. B1. A few sentences have been added here for clarity.

During this experiment, the source water flow in the PTR-MS was kept constant, and the  zero air measurement has been subtracted  remove the contribution to the VOC signals from the PTR-MS source water reservoir.  Variations in the measured background are thus due to the sample humidity alone. These results suggest that using zero air (e.g. bypassing the SFCE) as the background could lead to overestimations of dissolved DMS and acetaldehyde measurements under similar PTR-MS settings.

*Line 651: I suggest calling Appendix E: "Compilation of published solubilities for methanol, acetone, and acetaldehyde".*

Suggestion accepted.

Table 3: (and throughout). For consistency, I suggest sticking with nmol/dm-3 throughout and using scientific notation for isoprene instead of pmol dm-3. i.e. 9.96 * 10-3 +/-1.25 * 10-3 nmol dm-3)

      Suggestion accepted.

Table 2: Precision and limit of the detection of the seawater VOC measurements (6 min average).

| | measurement precision $1\sigma$ | Limit of detection |
|---|---|---|
| methanol (nmol dm$^{-3}$) | 6.52 | 19.56 |
| acetaldehyde (nmol dm$^{-3}$) | 0.17 | 0.51 |
| acetone (nmol dm$^{-3}$) | 0.44 | 1.32 |
| DMS (nmol dm$^{-3}$) | 0.0069 | 0.0207 |
| isoprene (pmol dm$^{-3}$) | 0.58 *10$^{-3}$ | 1.74 *10$^{-3}$ |
| benzene (nmol dm$^{-3}$) | 0.0043 | 0.0129 |
| toluene (nmol dm$^{-3}$) | 0.0042 | 0.0126 |

*Figure 3 (a): Typo "assuming" not "assuning".*

      Suggestion accepted.

[Figure]

*Figure 4: I suggest either using all black or using some color variation. It's hard to tell the 1:1 line and the fit to the measurement lines.*

Suggestion accepted. The figure has been changed to all black and the dash size of the fit to the measurement has been increased to make it easier to distinguish from the 1:1 line.

[Figure]

Figure 4: Invasion calibration curves for benzene (a), toluene (b), DMS (c) and isoprene (d) where a known amount of standard gas is added to the zero air carrier gas while measuring VOC-free Milli-Q water. Error bars were too small to display, but the noise associated with the measurement was found to be 0.0084 and 0.0044 ppbv for DMS and Isoprene respectively and 0.015 and 0.013 ppbv for benzene and toluene respectively. This was calculated as the std. dev. of 10 consecutive water blank measurements. A 1:1 line is included in 4 to illustrate the role of the water phase in absorbing these compounds.

*Figure 5: This plot and caption could use some clarification. This is a comparison of range solubilities observed with the SFCE-PTR-MS system and values predicted from literature. The meaning of the numbers in the legends (1-44) of Figure 5 are not immediately clear. To help, each line could be "Ref. x" (x=1-44), with "This work" as the thick red line and "S. P. Sander" as the thick blue one. In the caption, please write what you want the reader to take away from this demonstration. It seems like you are seeing lower solubility than the literature values.*

Suggestion accepted. The figure legend and figure description have been modified according to the reviewer's comments.

[Figure]

**Figure 4**: Evasion calibrations of OVOCs.  Displayed are the average experimentally determined slopes of 14 calibration curves of methanol (a) and acetone (b) and 11 calibration curves of acetaldehyde. These calibrations suggest possibly **lower** solubility of these compounds  compared to literature values. (c). Shaded area indicates one sigma standard deviation of the variance in the slope during this three-month period. Average experimentally determined calibration slope for methanol, acetone and acetaldehyde were 0.00786 ±0.00115 ppbv nmol$^{-1}$ dm$^3$, 0.0469 ±0.0145 ppbv nmol$^{-1}$ dm$^3$ and 0.0743 ±0.0190 ppbv nmol$^{-1}$ dm$^3$. Plotted along this are the predicted slopes using all experimentally determined solubilities as listed in R. Sander (2015). The recommended solubility by S. P. Sander et al., (2015) is plotted as a solid thick line in dark blue. The key to the figure is listed in the appendix, listing an in-text reference followed by the dimensionless water over air Henry solubility and the predicted slope using the experimentally determined solubility. For full reference of the cited solubilities, please refer to R. Sander (2015).

*Figure 7: Can this be remade as a full page plot? The horizontal axis is extremely tight.*

*If size is an issue, plot gases of similar magnitude on the same subplot and use the right axis. I suggest adding to the x axis "Sample Date & Time ( HH:MM DD/MM/YYYY)".*

Comments applied with thanks. To address the reviewer's comments, the horizontal axis range has been reduced to make it appear less tight. Additionally, the figure size was increased by approx. 30%. The horizontal axis has been labelled accordingly.

[Figure]

**Figure 7:** Selection of VOC measurements made from the ship's build in underway surface water supply (open symbols) and discrete samples from 5m rosette (closed symbols). The dotted line represents the limit of detection.

*Figure B-1: Are there error bars (like the caption says) in these plots? They are not rendering on my printer or pdf.*

Suggestion accepted.

[Figure]

**Figure B1:** Background dependence of DMS (a) and acetaldehyde (b) signal on the humidity in the sample air. Error bars represent the standard deviation of ten consecutive blanks.

---

## Author Comment (AC2) · 10 May 2019

**Reply to Reviewer 2 comments for: Segmented flow coil equilibrator coupled to a Proton Transfer Reaction Mass Spectrometer for measurements of a broad range of Volatile Organic Compounds in seawater**

Many thanks for the thoughtful comments from this anonymous reviewer. Please see our responses below. Reviewer comments are in italic and replies in normal font. The changes to the manuscript are presented as figures taken from the manuscript with the changes made indicated by red track changes.

*References: Please check the form of references shown in the text. For example, "Blando & Turpin, 2000" (page 1, line 28) should be shown as "Blando and Turpin, 2000", and "de Bruyn, Clark, Senstad, Barashy, & Hok, 2017" (page 2, line 43) should be shown as "de Bruyn et al., 2017".*

Suggestion accepted.

*Page 2, line 35 Please add the suitable references.*

Suggestion accepted.

Current estimates of air–sea VOC fluxes and the cycling of VOCs in the oceans have been limited in part by our ability to measure these compounds in  surface seawater . For example global budgets for acetone highlight the uncertainty of oceanic emissions (Fischer et al., 2012). A more recent sensitivity analysis  stresses the importance of accurate oceanic mixed layer concentrations on the global acetone budget, especially in the Southern Hemisphere (Brewer et al., 2017).

*Page 3, line 88 The response time of 10 min is only for isoprene, not for the other compounds such as acetone, methanol and so on.*

Suggestion accepted.

to PTR-MS to measure four different VOCs at a time (Williams et al., 2004). A bubbling-type equilibrator has also been developed for underway measurements of a range of dissolved VOCs with PTR-MS (Kameyama et al., 2010). The large volume of the bubbling equilibrator (i.d. 15.2 cm, height 100 cm) makes it very bulky and

90 creates a long response time (up to 18-19 min e.g. for methanol). Moreover, the high-water flow requirement of this type of equilibrator (1 dm³ min⁻¹) is also less suitable for discrete measurements.

*Page 3, line93-102 Those explanations itself are generally well, but not suitable in introduction section. I assume that those sentences could be removed.*

Suggestion accepted. This explanation has been moved to section 2.1.

The design of our SFCE is shown in Fig. 1. The SFCE is coupled to PTR-MS for measurement of methanol, acetone (2-propanone), acetaldehyde (ethanal), dimethyl sulphide (DMS), isoprene (2-methyl-1,3-butadiene), benzene and toluene (methyl benzene). These gases cover a large  range of solubilities (see Sect. 4.2.1). ,  demonstrating the versatility of the SFCE. The main advantage of this equilibrator lies in its design. Briefly, the segmented flow allows for a large surface area for gas exchange, ample equilibration time, and so a high degree of equilibration. The simple headspace and water separation system allows for rapid drainage of the sampled water as well as separation of the headspace from water without spray or droplet formation. This enables a fast response time . Due to the ease of changing the water sample intake and low water flow , the equilibrator can conveniently be used for both continuous underway and discrete measurements. The equilibrator is entirely made up of commercially available Polytetrafluoroethylene (PTFE) tubing and fittings, which should minimise adsorptive loss and make

4

the equilibrator relatively inexpensive and easy to replicate. The constant flow of water and smooth surfaces should also reduce bio fouling and facilitate occasional cleaning.

*Page 10, line 334 Ionized toluene should be found in not only m/z 93 but also m/z 79 depending on the drift tube voltage. Did the authors find the fragmentation of toluene? m/z 79 is used to identify benzene amount, therefore, the authors need to care of existence of the fragmentation.*

Suggestion accepted. A sentence highlighting this uncertainty has been added to the manuscript.

fragment ion m/z 41 and 39 respectively. This is in general agreement with Schwarz et al. (2009). This fragmentation ratio increases with increasing drift tube voltage (see appendix). It is possible that some of the mass 79 measured here contains a contribution from fragmenting toluene. However, because the gas standard contains both it is not straightforward to evaluate the magnitude of this interference.

*Page 11, line 354 What is the ratio of "100 cm3n:100 cm3" here? Maybe typo?*

Suggestion accepted. The "n" here is used to emphasized that this is a normalized mass flow delivered by a mass flow controller. This has been highlighted on page 4, line 115-116.